

# Simulating the roles of crevasse routing of surface water and basal friction on the surge evolution of Basin 3, Austfonna ice-cap

Yongmei Gong[1], Thomas Zwinger[2], Jan Åström[2], Bas Altena[3], Thomas Schellenberger[3], Rupert Gladstone[4], John C. Moore[5]

[1]Disivion of Atmospheric Sciences, University of Helsinki, Helsinki, 00560, Finland

[2]CSC – IT Center for Science Ltd., Espoo, 02101, Finland

[3]Department of Geosciences, University of Oslo, Oslo, 0371, Norway

[4]Arctic Center, University of Lapland, Rovaniemi, 96100, Finland

[5]College of Global Change and Earth System Science, Beijing Normal University, Beijing, 100875, P.R. China

*Correspondence to*: Yongmei Gong (yongmei.gong@helsinki.fi)

**Abstract.** The marine-terminating outlet in Basin 3, Austfonna ice-cap has been accelerating since the mid-1990s. Step-wise multiannual acceleration associated with seasonal summer speed-up events was observed before the outlet enters the basin-wide surge in autumn 2012. We use multiple numerical models to explore hydrologic activation mechanisms for the surge behavior. A continuum ice dynamic model is used to invert basal friction coefficient distributions using the control method and observed surface velocity data between April 2012 and July 2014. This provides an input to a discrete element model capable of simulating individual crevasses, with the aim of finding locations where summer melt water enters the glacier and reaches the bed. The possible flow paths of input surface melt water at the glacier bed and basal melt water are calculated according to the gradient of the hydraulic potential.

The inverted friction coefficients show the 'unplugging' of the stagnant ice front and expansion of low friction regions before the surge reaches its peak velocity in January 2013. Crevasse distribution reflects to a high degree the basal friction pattern. The melt water reaches the bed through the crevasses located above the margins of the sub-glacial valley and the basal melt that is generated mainly by frictional heating flows either to the fast flowing units or potentially gets accumulated in an over-deepened region. Based on these results, the mechanisms facilitated by basal melt water production, surface melt water and crevasse opening, for the surge in Basin 3 are discussed.

## 1 Introduction

Austfonna ice-cap, located on Nordaustlandet in the Svalbard archipelago, is the largest ice mass in the Eurasian Arctic in terms of area (7800 km$^2$) (Moholdt and Kääb, 2012). Basin 3 is one of its southeastern basins containing marine-terminating outlet glacier. The glacier is largely marine-grounded to as much as 150 m below sea level and is known to have surged around 1850-1870 (Dowdeswell et al., 1986).

The northern flow unit of the outlet glacier experienced long-term acceleration since the mid-1990s (Dowdeswell et al., 1986) along with stepwise inter-annual accelerations since 2008. These short-lived summer speed-up events occur during the surface melt season (Dunse et al., 2015). The southern corner of Basin 3 accelerated to about 290 m a$^{-1}$ in spring 2008 but had decelerated again by spring 2011 (Gladstone et al., 2014). However high velocities were again observed in the same area during spring 2012 which subsequently gradually increased to ~1800 m a$^{-1}$ after the summer melt season and before a basin-



wide surge took place in autumn 2012 (Dunse et al., 2015). The surge reached its peak in January 2013 with a maximum velocity of ~6500 m a$^{-1}$.

The 130-140 year long quiescent phase of Basin 3 is similar to other Svalbard glaciers, but the two decades long accelerating phase of the northern flow unit exceeds those of other glaciers such as the 7-11 years of Monacobreen (Strozzi et al., 2002). The step-wise multi-annual acceleration observed since 2008, associated with seasonal summer speed-up events, is also

exceptional from other surging glaciers in Svalbard. Similar melt season speed-up events have been observed in Greenland, and provides evidence for rapid, large-scale, dynamic responses of the ice sheet to climate warming (Sundal et al., 2011; van de Wal et al., 2008; Zwally et al., 2002). Sundal et al (2011) pointed out that a simple model of basal lubrication alone cannot simulate the fast flowing manner of the glaciers on Greenland ice sheet, and that an improved understanding of sub-glacial drainage would be essential for model studies to capture ice dynamic responses to climate warming. This applies also to the

surge in Basin 3, which requires a mechanism involving both thermal and hydrologic changes to explain the inter-annual and seasonal accelerations (e.g. Dunse et al., 2015; Gladstone et al., 2014).

The glacier in Basin 3 (recently named Storisstraumen) is polythermal, with a maximum ice thickness of 567 m, sufficient to raise internal ice temperatures to the pressure melting point (pmp) (Dunse et al., 2011). Where the ice is thinner, closer to the margins, the ice is probably frozen to the bed except under fast flowing outlets. In principle the surge of polythermal glaciers

can be explained by a soft-bed mechanism with some constraints for the initiation, such as the unfreezing of the cold bed by the evolution of the thermal regime or by the input of melt water from englacial channels (Clark, 1976; Lingle and Fatland, 2003; Robin, 1955).

Gladstone et al. (2014) suggest soft-bed sliding mechanisms involving feedbacks in the hydrological system at the ice-till interface responding to penetration of surface melt to explain the summer speed-up events observed since 2008. Surface

meltwater can penetrate cold and polythermal glacier ice in High Arctic glaciers and the Greenland ice sheet through moulins and fractures that cut down all the way to the glacier bed (e.g. Benn et al., 2009; Copland et al., 2003; van de Wal et al., 2008; Zwally et al., 2002). Water-filled crevasses can penetrate to the glacier bed regardless of ice thickness or crevasse spacing as long as the tensile stress acting normal to the crevasse exceeds about 100kPa (Boon and Sharp, 2003; van der Veen, 1998). Bougamont et al (2014) investigated the sensitivity of the basal hydrology system in the Russell glacier catchment to the

volume of surface melt delivered to the bed, finding increases in surface melt volumes lead to faster summer flow.

Dunse et al. (2015) has suggested a "hydro-thermodynamic" feedback whereby summer meltwater penetrating to the bed is not considered a purely external forcing to the system: meltwater penetrating crevasses to reach the bed enhances basal processes such as lubrication and sediment deformation resulting in enhanced ice flow and potentially an increase in extensional stress, which may in turn cause increased crevasse formation over a wider area, routing more melt water down to

the bed.

These earlier studies highlight the importance of time-evolving basal temperature and friction, which are strongly influenced by the evolution of a basal hydrology system. The basal hydrology system can be fed both by in situ melting and by surface meltwater, and has the capacity to not only directly cause sliding but also to alter the thermal regime and hence deformational flow.

Previous studies of modeling crevasse simulate the formation of fractures as a continuous process. They treat the development of cracks on a macroscopic scale by either using simplified parameterization of fracturing effects via variables such as depth



of crevasse (Cook et al., 2014; Nick et al., 2010, 2013; Weertman, 1973) or using Continuous Damage Mechanics (CDM), which simulates the continuous process from micro-scale cracks to macro-scale crevasses (Albrecht and Levermann, 2014; Bassis and Ma, 2015; Borstad et al., 2012, 2016; Krug et al., 2014). In this study we take a different approach and apply a

first-principles discrete element model (Åström et al., 2013, 2014) capable of simulating crevasse formation as a microscopic scale discrete process in addition to the continuum ice dynamics models. The discrete element model (HiDEM) is used to determine the locations of the crevasses penetrating though the full thickness of the glacier whereby surface water may reach the bed.

In Sect. 2 we present the observational data used for setting up the simulations and validating the results. In Sect. 3 we present

the methodology. We use a continuum ice dynamic model to invert the basal friction field from approximately monthly observations of ice surface velocities between April 2012 and July 2014. This basal friction field then acts as a boundary condition for basal sliding in our discrete element model that simulates crevasse distribution in the lower part of Basin 3 for particular points in time. In Sect. 4.1 we firstly investigate the evolution of the basal conditions in Basin 3 during and after the peak of the surge. In sect. 4.2 we present the modeled crevasse distributions before and after maximum surge velocity and

validate the latter with crevasse map derived from satellite imagery. In Sect. 4.3 we locate the crevasses that reach the bed, calculate basal melt rates and estimate the flow path of the basal water. In Sect. 5 we discuss the mechanisms facilitated by basal melt water production, surface melt water and crevasse opening for the surge occurred in Basin 3.

## 2 Observations

### 2.1 Surface and bedrock topography data

Surface elevation was derived from Cryosat altimetry data acquired during July 2010 – December 2012 (McMillan et al., 2014). McMillan et al., (2014) grouped measurements acquired over a succession of orbit cycles that are within 2-5 km$^2$ geographic regions. Bedrock elevation (Dunse, 2011) was derived by point-wise subtracting the measured ice thickness from a 250 m resolution surface elevation that is based on the Norwegian Polar Institute (NPI) 1:250 000 topographic maps derived from aerial photography over Austfonna in 1983. The ice thickness used for generating bedrock elevation was based on

airborne radio echo sounding (RES), (Dowdeswell et al., 1986) supplemented with two RES data sets from 2008 (Vasilenko et al., 2009). Marine bathymetry (2 km horizontal resolution) was from the International Bathymetry Chart of the Arctic Ocean, Version 2.0 (Jakobsson et al., 2008). Bathymetry and inland bedrock elevation were combined by using an interactive gridding scheme to eliminate the mismatch along the southern and northwest coast line (Dunse, 2011).

We point out several bed topography features (Fig. 1b) that are important to the investigation here. The sub-glacial hill located

at roughly 700 km E and 8850 km N rising to about 250 m a.s.l. A corresponding, but smaller magnitude, bedrock maximum exists at the opposite side of Basin 3, approx. 15-20 km southwest of the aforementioned hill. A subglacial valley runs between these bedrock maxima, and extends several tens of km upstream and downstream. The minimum bedrock height for Basin 3 is within an over-deepening in the lower part of the valley. The importance of these features is discussed in more detail in Sect. 4 and 5.

### 2.2 Surface velocity data



We used velocity time series maps (April 2012-July 2014) generated from TerraSAR-X (TSX) satellite synthetic aperture radar (SAR) scenes, (Table 1; Schellenberger et al., 2017, in review) as the input surface velocity data for basal friction coefficient inversion. These maps were based on original 2m resolution TSX scenes provided by the German Aerospace Center (DLR) covering only the lower part of Basin 3 (Fig. 1a). To generate the final velocity maps for the times between two successive TSX images, which were geocoded using a DEM of Austfonna (Moholdt and Kääb, 2012), we needed to use displacement maps. The displacement maps between two consecutive acquisitions were determined using cross-correlation of the intensity images (Strozzi et al., 2002).

The coverage of the TSX velocity was smaller than the model domain used by our ice dynamic model (Fig. 1a). Therefore we stitched the TSX data on top of two background velocity fields with larger coverage depending on the acquiring time. The TSX data derived during 19 April 2012 – 28 December 2012 was stitched with velocity snapshot from ERS-2 (European Remote Sensing Satellite 2) SAR observation acquired in March to April 2011(Gladstone et al., 2014; Schäfer et al., 2014); and the TSX data derived after 28 December 2012 was stitched with velocity snapshot from Landsat-8 imagery acquired in April 2013. We then applied a row-wise recalculation of the velocity value for the grid points on the model mesh that were upstream from the TSX velocity map coverage (Fig. 1a) to create a smoother transition from TSX velocity map to the background velocity map. The recalculation was carried out by weighting the background velocity data and TSX velocity data according to the distance between the column indices of the targeting grid point and the column indices of the first grid point that had value from TSX velocity map on the same row.

The velocity recalculated for the upstream area is simply to avoid numerical instability that might appear at the boundary between the TSX and background velocities. So as not to bias the crevasses distribution calculation, we confined the discrete element model domain to a smaller region close to the ice front, which was fully covered by TSX velocity map and far away from this transition zone (Fig. 1a).

### 2.3 Crevasse map

We created a crevasse map from satellite imagery to validate our modeled crevasse distribution. The map was generated from a Landsat 8 image acquired on 4th August 2013 using the Radon-transform technique (Petrou and Kadyrov, 2004; Toft and Sørensen, 1996). We experimented with crevasse maps created from various different satellite sensors (Landsat 7, Landsat 8, ASTER, Sentinel-2), but here we used only the Landsat 8 scene which combines good spatial coverage with high radiometric quality.

The Radon-transform has been demonstrated to be efficient in detecting along flow features (Roberts et al., 2013), but can also be used for complex flow patterns, like the one in Basin 3 which has a wide range of crevasse orientations. The advantages of the Radon-transform over other detecting methods are that crevasse patterns can be extracted where edge detectors methods (Bhardwaj et al., 2015; Wesche et al., 2013) would fail, and also that it is more robust than frequency-domain methods (Sangwine and Thornton, 1998) in detecting crevasses from incomplete coverage due to cloud coverage, image borders or the calving front.

In this study we followed a similar approach as Roberts et al. (2013), but used a more robust implementation and a different post-processing procedure. Firstly, the satellite image was pre-processed with a Laplacian-filter to prioritize the high frequencies, e.g. to sharpen the edges of surface features. We performed the Radon-transform, $R(p, \theta)$ on 300 m×300 m



subsets of the satellite image, and project the image intensity $I(x, y)$ along lines with tangent vectors oriented at $\theta$ to the x-axis and offset by a perpendicular distance, $p$, from the origin (Toft and Sørensen, 1996):

$$R(p,\theta) = \int_{-\infty}^{\infty} \int_{-\infty}^{\infty} I(x,y)\delta(-x\,sin\theta + y\,cos\theta - p)dxdy, \tag{1}$$

where the 2D integration is restricted by the Dirac delta function, $\delta(-x\,sin\theta + y\,cos\theta - p)$, to the appropriate straight line in the x-y plane. The range of the transform coordinates is a half circle ($0 \leq \theta < \pi$) and $p$ is the spatial integral ranging over the domain of the subset of the image. The result of the transform was a 2D feature space at different azimuthal orientations ($\theta$). To capture both small and big crevasses, we re-sampled the image intensity $I(x, y)$ in each 300m×300m image subset with a resolution of 2 pixels and again implement a weighted Radon-transform function, where a mask over the subset was used to

remove features like image borders, clouds, ocean etc. The resulting Radon transformation of a subset was again a two dimensional subset. Then the standard deviation at each orientation was used to extract the response for elongated texture:

$$s(\theta) = \sqrt{\frac{\sum_{i=-P}^{P}\left(R(i,\theta) - \bar{R}(\theta)\right)^2}{N+1}} \tag{2}$$

Here the overbar denotes the mean and $N$ denotes the amount of steps within the domain of $p$. Finally, a running median filter with a spacing of two ($\varDelta = 1°$) was used to remove noise:

$$\tilde{s}(\theta) = median\{s(\theta - \Delta), \cdots, s(\theta + \Delta))\} \tag{3}$$

The results of the procedure were maps showing the dominating azimuthal orientations ($\theta$) of the crevasse clusters (Fig. 2a) and their response ($\tilde{s}(\theta)$) (Fig. 2b) in each 300 m×300 m window. To use the detected crevasse zones as a validation for our modeled crevasse distribution we transformed $\tilde{s}(\theta)$ and the orientation ($\theta$) into a cartographic representation (Fig. 2c). To do so, an empty image was randomly seeded with high intensities. Then a kernel with an elongated shape was convoluted over

the image. This kernel was adaptive, as the orientation of the elongated shape is dependent on the orientation of the highest responding orientation signals in every window.

## 3 Methodology

### 3.1 Basal friction inversion in the ice flow model

The continuum ice dynamic model we used is Elmer/Ice, an open-source finite element model for computational glaciology

(Gagliardini et al., 2013). In this study, the simulations with Elmer/Ice were carried out by considering a gravity-driven flow of incompressible and non-linearly viscous ice flowing over a rigid bed. The constitutive relation for ice rheology was given by Glen's flow law (Glen, 1955), and the ice flow was computed by solving the Stokes equations. A fixed calving front criterion was adopted in all the simulations in this study due to the lack of ice thickness information corresponding to the observed calving front positions after 2011. The criterion assumes that the calving front was always grounded with a positive

height above floatation, which re-flects the observation at the terminus in Basin 3. As the frontal and near-frontal region are not confined between lateral walls we would not expect significant impact of different calving front positions on longitudinal stress gradient upstream, i.e. the migration of calving front may have less impact on the basal shear stress distribution in the upstream area than the uncertainties brought by the observed ice velocity or the lack of ice thickness information at the calving





front. On the other hand the basal shear stress calculation at the ice terminus will be effected. However the glacier bed is

already very 'slippery' at the ice terminus. And as the ice front in the simulation did not advance the calving flux might be

underestimated.

We performed inverse modeling of basal friction coefficient distributions from surface velocity observations using Elmer/Ice

based on the control method (MacAyeal, 1993; Morlighem et al., 2010), and implemented in Elmer/Ice by Gillet-Chaulet et

al (2012). The inverse modeling determines the spatial distribution of the basal friction coefficient, $C$ defined by:

$$\tau_b = C\, u_b, \tag{4}$$

where $\tau_b$ is the basal shear stress and $u_b$ is the basal velocity, by minimizing the mismatch between modeled and observed

surface velocity as defined by a cost function:

$$J_o = \int_{\Gamma_s} \frac{1}{2}(|u_{mod}| - |u_{obs}|)^2\, d\Gamma, \tag{5}$$

where $|u_{mod}|$ and $|u_{obs}|$ are the magnitude of the modeled and observed horizontal surface velocities. The mismatch in the

direction of the velocity components is ignored. And only a match of velocity magnitude is optimized.

All simulations were computed on an unstructured mesh over Basin 3 generated with the open source software GMSH

(Geuzaine and Remacle, 2009). The element size of the mesh increased from ~150 m at the glacier terminus to 2500 m at the

back of the basin. The 2D mesh was then vertically extruded between the interpolated bedrock and surface elevation into 10

equally spaced terrain-following layers to form a three-dimensional (3D) mesh.

All inversions were done sequentially in chronological order with a transient simulation after each inversion to evolve the

geometry for the next inversion. Temporally and spatially varied ice temperatures in the flow solution were accommodated

using an iterative process detailed in Gong et al. (2016). A month of geometry evolution starts with the $C$ field inverted

from the first velocity map acquired during that month to evolve the glacial geometry for 30 days with temporal resolution

of half a day, and mean 1990-2000 surface mass budget (SMB) forcing from the regional climate model HIRHAM 5

(Christensen et al., 2007). In the case of acquisition time gaps (Table 1; mostly after August 2013) transient simulations

were carried out for the length of the gap with the latest $C$ distribution and temporal resolution of one day.

### 3.2 Crevasse distribution calculation by a discrete element model

HiDEM is a first-principle model for fracture formation and dynamics. The version of HiDEM we used here is a purely elastic

model, rather than incorporating visco-elastic processes (Åström et al., 2013). This version is sufficient for the purposes of

locating fractures given only geometric boundary conditions and basal friction coefficient. If the initial condition of HiDEM

is out of equilibrium, elastic deformation within the ice will appear as a result of Newtonian dynamics. If local stress exceed

a fracture criterion, the ice will begin to rupture. Compared to viscous deformation, elastic deformation is very rapid, and

typically a glacier will approach a new equilibrium after some minutes of simulated dynamics. Consequently, at the end of a

simulation, a crevasse field has been formed. A detailed description of HiDEM can be found in Åström et al. (2013, 2014).

The simulations were set up with input data from marine bathymetry, bedrock topography, $C$ field, and the surface topography.

We selected two $C$ snapshots inverted from velocity data observed in 18-29 August 2012 ($C_{pre}$) and 16-27 August 2013 ($C_{post}$)

(Fig. 3) as boundary condition for basal sliding in HiDEM. Those dates were chosen to model the crevasse distribution after





the summer melt season before and after the peak in surge velocities observed in January 2013. All the simulations were carried out with 50 m spatial resolution for crack separation and $10^{-4}$ s time step. The computations were carried out on an

HPC cluster using typically 500 computing cores for a few hours.

## 4 Results

### 4.1 Basal friction evolution

We investigate the evolution of basal friction using inverse modeling to determine $C$ from the observed surface velocity between April 2012 and July 2014, spanning the period of the Basin 3 peak surge velocities in January 2013. We focus on the

lower region close to the terminus that is fully covered by TSX velocity observations.

To make the pattern of the $C$ distribution clearer we plotted the common logarithm of $C$ ($\log_{10}(C)$), instead of $C$ itself. Figure 4a shows a clear expansion of low friction area ($\log_{10}(C) \leq$ -3.5) both inland and to the frontal region in the southern basin before the glacier enters the peak of the surge. In 2011 the low friction patches in the central and southern basin were disconnected from the inland region and also behind a stagnant terminus.

In April 2012, before the summer melt season, a low friction region also appeared in the southern corner, though still with a stagnant ice front. The low friction area of the northern flow unit slightly expanded to the south through the relatively flat frontal area. However, the fast flow did not expand beyond the margin of the sub-glacial valley, which exited through the northern part of the calving front (Sect. 2.1; Fig. 4a), and might impose some restriction to the expansion of fast flow. After the summer melt season (August 2012) the stagnant frontal region shrank to the glacial terminus which might have thinned

to reach a condition close to floatation (McMillan et al., 2014). During this period the low friction area underneath the southern part expanded further inland and became connected to the northern low friction area. In January 2013 the glacier entered a basin-wise surge and the low friction area also expanded across the entire width of the basin near the calving front with a few particularly deep minima ($\log_{10}(C) \leq$ -5.5; almost vanishing friction) in the south (Fig. 4b).

After January 2013 the basal friction pattern in northern basin remained almost stable. The almost vanishing friction area

($\log_{10}(C) \leq$ -5.5) in the southern frontal region gradually shrank back inland away from the terminus.

### 4.2 Crevasse distribution and validation

We used a minimum fracture width of 0.05 m to identify a crevasse in HiDEM, which allowed us to keep most of the fractures across the whole model domain. Many fractures were generated upstream of the sub-glacial hill (the area inside the black box in Fig. 5b); these were caused by boundary effects due to the limited domain and are excluded from the study (illustrated by

the region in Fig. 1a). A similar boundary effect causes incorrect crevassing in the southwest corner of the domain (also marked in black in Fig. 5b). All of these artificial crevasses are irrelevant to the water routing and surge processes we focus on in this paper. We defined cut-through crevasses as crevasses that penetrate through 2/3 ice depth and assume that they could cut through the full depth of ice if filled with water and potentially route surface melt water into the basal hydrology system vertically.

The crevasse distribution from $C_{post}$ was validated using the crevasse map generated from satellite observations acquired on 4[th] August 2013. The cartographic map of the crevasse detection (Fig. 2c) from the satellite observation was used for the



validation. To estimate the statistical quality of the simulated crevasse field with the observationally estimated map we calculated the Kappa coefficient ($K$) (Wang et al., 2016). As almost any two maps will be significantly different with large sample size (> 62483) (Monserud and Leemans, 1992), we firstly re-sampled the two maps to an appropriate resolution.

Experimentation leaded us to require a 4.6×4.6 km smoothing window to achieve substantial agreement ($K = 0.71$) (Cohen, 1960) between the maps. At higher resolutions $K$ is worse for a variety of reasons: the ice dynamics model cannot advect crevasses, hence many crevasses in the image that in reality were created further upstream were simply not present in the simulation; crevasse densities are very variable and even at 1.5 km resolution the distribution is not smooth ($K = 0.45$); and the observationally derived map is not a perfect representation of reality. We next discuss the crevasse patterns derived from

observations and those from the discrete element model in detail.

The crevasse map created by the Radon-transform shows a highly crevassed glacial lower region, which comprise sections with crevasses of different orientation (Fig. 2). Transverse crevasses that are almost perpendicular to the flow direction can be found in both northern and southern flow units, reflecting large longitudinal tensile stress after dramatic acceleration. However, the detection intensity of the crevasse in the northern flow unit is rather weak. The terminus between the northern

and middle flow units has a mixture of crevasses with orientations perpendicular to each other indicating the expansion and merging of the two flow units. Marginal crevasses can be found above the sub-glacial valley margins parallel to the local flow direction reflecting lateral shear stresses and longitudinal compressive stress caused by the presence of the valley margins.

The modeled crevasse distribution reflects the broad features the basal friction pattern (Fig. 5b). A high crevasse density is generated in areas with large tensile stress caused by extending flow on the lower part of basin 3, as well as at shear margins

between low and high friction areas. The orientation of the modeled crevasses above the sub-glacial valley margins agrees with the observation (Fig. 5b). However orientations of most of the modeled crevasses in the middle upper area have a ~60° mismatch with the satellite image (Fig, 5c) and the modeled crevasse density at the frontal area of the southern and northern flow units are larger than those in the observationally derived map.

This mismatch of the orientation between the modeled and observationally derived crevasse distribution in the middle upper

area (Fig. 5c) may be due to HiDEM only simulating the ad-hoc formation and not advection of crevasses, thus no crevasse formation history can be inferred from the model. The inclusion of crevasse advection could be implemented in a two-way coupling of HiDEM with a continuum model in future studies. The mismatch of the crevasse density (Fig. 5c) at the northern and southern frontal area could be caused by the mismatch of ice front position between the reality and the model. Although in reality the ice front advanced for several kilometers after the full-surge, it was kept fixed in position in Elmer/Ice (Sect.

3.1). The shape and steepness of the ice front likely affects the behavior of the discrete element model. However, as they are concentrated at the terminus of the glacier, these crevasses are less likely to affect the basal hydrology system on a wider scale.

To investigate the crevasse distribution after the summer melt season in 2012, we used $C_{pre}$ and the corresponding geometry with HiDEM. The configuration produced more crevasses in the frontal region of the northern flow unit than in the southern

flow unit and almost no crevasses over the frontal region of the central flow unit (white dots in Fig. 5a). Crevasses also appeared at the margins of the sub-glacial valley.

By looking at the overall crevasse distributions in August 2012 and August 2013 (white dots in Fig. 5a and 5b) together with their corresponding $C$ distributions (Fig. 4) we noticed that the outline of the densely crevassed region more or less follows




the outline of the low friction region, indicating the governing role of basal friction on crevasse formation. This was also shown by the fact that there were more crevasses formed in the southern and middle frontal area in August 2013 than in August 2012 as the bed was more 'slippery' in August 2013 (Fig. 4b). The confining effects of the bed rock topography to the fast flow, basal friction and crevasse distribution also became more visible in the later stage of the surge: the modeled crevasses at the sub-glacial valley's sides indicated a sharper boundary in August 2013 than in August 2012.

### 4.3 Surface and basal water sources

We selected the cut-through crevasses in August 2012 and August 2013 (red dots in Fig. 5a and b) to identify possible routes of surface water to the bedrock. In August 2012 most of the crevasses in the frontal area cut through the ice deep enough and very likely represent future calving locations for the terminus during its advance. Most of the crevasses located between the northern and southern fast flowing regions were shallow, surface crevasses. Many crevasses above the margins of the sub-glacial valley could reach the bed and potentially route surface melt water from upstream to the bed. By August 2013 more

cut-through crevasses had been developed in the lower southern and central basin compared with August 2012 as velocity gradients significantly increased after the basin-wide acceleration. There were more cut-through crevasses present above the shear margin but almost no cut-through crevasses above the over-deepening area.

Using the locations of cut-through crevasses above the margins of the sub-glacial valley that could potentially route surface melt water down to the bed, in August 2012 we calculated the sub-glacial water flow path according to the gradient of the

hydraulic potential (Fig. 6a). The hydraulic potential ($h$) was calculated as below:

$$h = (z_s - z_b)\frac{\rho_i}{\rho_w} + z_b \tag{6}$$

in which $z_s$ and $z_b$ are surface and bedrock elevation; $\rho_i$ = 910 kg m$^{-3}$ and $\rho_w$ = 1000 kg m$^{-3}$ are the density of ice and water.

The flow paths are generated by integrating through the vector field that follows the steepest descent in $h$ using fourth-order Runge-Kutta method.

The surface melt water entering the bed at the north was predicted to either flow directly to the terminus, or stop at the sub-glacial over-deepening area (Sect. 2.1; Fig. 6a). Surface melt water entering the bed from the south was routed directly towards the terminus at the southern corner of the glacier, suggesting that surface melt contributed to the dramatic acceleration of the southern part of Basin 3 after the summer melt season in 2012.

In addition to the basal water supplied via the crevasse system, we also estimated the basal melt rate (Fig. 6b) for the temperate

base area of the glacier. Within Elmer/Ice we computed the energy-balance at the bed from an estimated geothermal heat flux, strain heating and basal friction-heating (Gong et al., 2017). Relatively high basal melt rates (> 0.005 m a$^{-1}$) appeared at the side walls of the sub-glacial valley around the over-deepening area, mainly caused by frictional heating. The basal melt water followed similar patterns of flow as the surface melt water that reaches the bed.

### 5 Discussion

Previous studies of the surge in Basin 3 (Dunse et al., 2012, 2015; Gladstone et al., 2014) revealed an atypical surge activation phase with multi-decadal acceleration superimposed, for at least 6 years, by short-lived, abrupt seasonal speed-up events that





were clearly related to summer melt., which could not be explained solely by the thermal switch mechanism (Murray et al., 2003) typical of polythermal surging glaciers in Svalbard.

We used the discrete element model – HiDEM (Åström et al., 2014) to locate the possible location of crevasse factures that may penetrate ice deep enough to act as routing-paths of surface melt water to the bed. In this study we focused on the cut-through crevasses formed above the margins of the sub-glacial valley because the basal flow pattern of the surface melt entering through those crevasses was indicative of potential subglacial water routing and hydrology.

We agree that the so-called "hydro-thermodynamic" feedback proposed by Dunse et al. (2015) could explain the development of the surge in Basin 3 in general. Based on our results we now further present arguments to emphasis the role of crevasse
formation, summer melt and basal hydrology system played in the seasonal speed-up events.

Firstly, our calculation of the flow paths of both surface melt entering through the crevasses and basal melt water production suggest the potential of a direct lubricating effect acted beneath the northern and southern fast flow units. Figure 6 shows that water entering through the crevasses downstream from the subglacial mountain (the flow paths in the northern half of Fig. 6a) will flow through the area where the northern fast ice flow unit has developed. The water accessing the bed at the southern
part of the basin travels directly towards the terminus at the southern corner of the glaciated system, which has dramatically accelerated during the melt season in 2012.

Secondly, some of the basal water flow paths presented in Fig. 6a and 6b terminate under a plateau in the hydraulic potential which occurs in the over-deepened bedrock region (see also Fig. 1b). Given the very low gradients of our calculated hydraulic potential in this region and the presence of a local hydraulic potential minimum slightly downstream of the over-deepening,
basal water would likely have low flow speeds, and possibly even accumulate in the over-deepened bedrock region, over time. This may have impacted on the surge development in Basin 3. Also given that the lowest basal resistance during most of 2012 (Fig. 4a) was immediately downstream of the over-deepening area in the northern flow unit, outflow of accumulated water likely enhanced the surge activation here. If seasonal surface melt water accumulates here and drains over a longer period, this may explain prolonged high ice velocities after the melt season has ended.

The temporary speed-up of the southern flow unit in 2008 (Gladstone et al., 2014) could plausibly have been triggered by an influx of basal water that was not repeated again until the basin wide surge was initiated. An outburst of basal water accumulated in the over-deepened bedrock region could provide one possible mechanism for this to occur. A ridge in hydraulic potential divides the northern and southern flow units in August 2012 (Fig. 6). An anomalously high inflow of surface meltwater could have caused this ridge to be flooded if regular drainage channels were of insufficient capacity. We are unable
to say how likely this is without a time series of surface melt data including the 2007 and 2008 seasons, but such an event could cause a temporary speed-up to the southern flow unit.

Englacial channels which may cause a redistribution of water within the hydrologic system (Fountain and Walder, 1998) are not directly considered in the current study. We assume that direct transfer of surface runoff via cut-through crevasses exceeds the englacial water transport at Basin 3.

Lastly, we look at the role of basal melt water in the activation of the southern flow unit. Basal meltwater from further upstream in the northern flow unit can drain toward the southern unit (Fig 6b; prior to the basin-wide surge, nearly all of the ice drained toward the northern flow unit). If this basal meltwater accumulated upstream due to the lower part of the glacier being below pressure melting point, such accumulated basal melt water could have caused the speed-up once basal temperatures reached melting point under the southern corner and the hydrologic system extended beneath the southern flow



unit. Also basal melt water generated locally in the over-deepening area (Fig. 6b) may not have been able to drain completely in one season thus could be accumulated locally. However whether basal melt water can eventually burst out from the over-deepening area and contribute to the seasonal speed-up events or refreeze locally depends also on the development of the hydrology system and the thermal regime.

Although we lack either simulated or observed surface melt volumes for summer 2012 we would expect that the surface melt

is much larger than basal melt. The runoff output from HIRHAM5 regional climate model in 1995 (personal information from R. Mottram of Danish Meteorological Institute; 1995 was not a year with high surface melt) at the location of the cut though crevasses was at least 10 time larger than the basal melt rate in either 1995 or 2012. The volume of surface melt observed at weather stations located in southwestern Basin 3 after summer in 2004 was also at least 10 times larger (Schuler et al., 2007). Considering the seasonal timings and magnitudes of speed-up events, and the feedback between surface melt water input and

hydraulic warming at the bed, it is clear that surface melt, when it can penetrate to the bed, causes a much higher impact on sliding and ice dynamics than basal melt water.

In the end, our results support the "hydro-thermodynamic' mechanism, in which crevasses provide access for surface melt water to reach the bed. We have demonstrated that cut-through crevasses are likely to be present approaching the surge in Basin-3, and that water flow paths route surface meltwater along flow paths corresponding to the regions of observed fast

flow. While Dunse et al. (2015) are unspecific as to the cause of "hydro-thermodynamic" initiation zone, we propose that basal melt water, resulting from the build-up of the reservoir area and gradual thickening of ice (and hence raising of basal temperatures) during the quiescent phase, could sufficiently enhance flow speeds to initiate cut-through crevasses. Given that basal meltwater fluxes are likely to be at least an order of magnitude lower than surface meltwater or runoff fluxes, their impact on glacier sliding is likely to be much smaller. We suggest that basal meltwater, which is likely to be primarily routed

toward the northern rather than southern flow unit due to topographic constraints (Fig. 6b), caused the speed up from the quiescent phase during the last part of the 20th century and early 21st century. This would require two key developments from quiescent to surge phase. Firstly, the initiation of sliding after ice thickening provided sufficient insulation for the bed to reach pressure melting temperature and generate sufficient meltwater, which could have occurred during the early nineties. Then at some point before August 2012 extensional flow due to sliding could have become sufficient to cause cut-through crevasses,

leading to further acceleration and the surge onset due to the annual "hydro-thermodynamic" feedback.

It is not clear at which point the "hydro-thermodynamic" feedback cut in, though it is likely to have first occurred in the northern flow unit, due to this unit's earlier acceleration. We suggest that the "hydro-thermodynamic" feedback cut in for the southern unit in 2011 or early 2012 due to crevasses penetrating near the southern margin (Fig. 5a), rapidly causing the basin wide surge.

**6 Conclusions**

We have forced the discrete element model HiDEM with outputs from the continuum ice dynamic model Elmer/Ice to study the crevasse distribution during the surge in Basin 3, Austfonna ice-cap in 2012-2013. Our continuum to discrete multi-model approach provides simulated locations where cut-through crevasses allow surface melt water to be routed to the bed. We have demonstrated that automatic crevasse detection through Radon-transform may be used to validate simulated crevasse

distribution from our continuum-discrete modelling approach. With the future addition of a basal hydrology model, the current




study constitutes a step towards a fully coupled ice-dynamic englacial/basal hydrology modelling system in which both input locations of input surface water and basal meltwater generation are represented.

Our results support the "hydro-thermodynamic" feedback to summer melt proposed by Dunse et al (2015) to explain the seasonal speed-up in Basin-3 and the initiation of the acceleration of the southern flow unit in 2012. The calculated flow paths

of the basal water according to hydraulic potential indicate either a direct enhancement to the ice flow through basal lubrication or a lagged-in-time mechanism through the outflow of accumulated water in the over-deepening area.

We propose that basal melt water production caused the speed up from the quiescent phase of Basin 3 during the last part of the 20th century and early 21st century. Then the "hydro-thermodynamic" feedback initiated during 2011 or early 2012 causing the activation of the southern flow unit and the expansion of the surge across the entire basin. The quantification of

the roles and mechanisms involving basal melt water production, surface melt water and crevasse opening for the surge discussed in this study need to be further improved by coupling basal hydrology with the thermal regime evolution and surface mass and energy balance.

*Author contribution.* Y. Gong and T. Zwinger designed the numerical experiments and carried out the simulation in Elmer/Ice. J. Åström carried out the simulations in HiDEM. B. Altena produced the crevasse map with Radon-transform. T.

Schellenberger processed and produced the TSX velocity time series. Y. Gong analyzed the model results and designed the figures. Y. Gong wrote the manuscript together with R. Gladstone, J. Moore and T. Zwinger. All the authors assisted in data interpretation and commented on/edited the paper.

*Acknowledgements.* We wish to thank all the partners for providing data and constructive discussion during the study, especially R. Mottram from Danish Meteorological Institute for the HIRHAM5 surface mass balance; T. Strozzi from

GAMMA Remote Sensing and Consulting AG for the ERS-2 SAR surface velocity observation acquired in March to April 2011; M. McMillan from the University of Leeds for the surface elevation derived from Cryosat altimetry data acquired during July 2010 – December 2012; T. Dunse from the University of Oslo for the bedrock and ice thickness data and A. Kääb from the German Aerospace Center DLR (LAN_0211) for TSX data. We wish to thank F. Gillet-Chaulet from Laboratoire de Glaciologie et Géophysique de l'Environnement for making available his code for the inverse modelling. We also

acknowledge CSC IT Center for Science Ltd. for the allocation of computational resources. The work was supported by Finnish Academy project 286587: Simulating Antarctic marine ice sheet stability and multi-century contributions to sea level rise. T. Zwinger is supported by the Nordic Center of Excellence eSTICC (eScience Tools for Investigating Climate Change in Northern High Latitudes) funded by Nordforsk (grant 57001). B. Altena is funded by the European Research Council under the European Union's Seventh Framework Programme grant agreement No. 320816. T. Schellenberger was funded by the

Research Council of Norway (RASTAR, 208013), the Norwegian Space Centre as part of European Space Agency's PRODEX program (C4000106033), and the European Union FP7 ERC project ICEMASS (320816).

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



Table 1 TerraSAR-X acquisitions of Basin 3 and repeat-pass period

| Repeat-pass period (days) | Start and end-date |
|---|---|
| 11 | 19 Apr 2012–30 Apr 2012 |
| 11 | 30 Apr 2012–11 May 2012 |
| 88 | 11 May 2012–7 Aug 2012 |
| 11 | 7 Aug 2012–18 Aug 2012 |
| 11 | 18 Aug 2012–29 Aug 2012 |
| 44 | 29 Aug 2012–12 Oct 2012 |
| 11 | 12 Oct 2012–23 Oct 2012 |
| 11 | 23 Oct 2012-3 Nov 2012 |
| 22 | 3 Nov 2012–25 Nov 2012 |
| 11 | 25 Nov 2012–6 Dec 2012 |
| 22 | 6 Dec 2012–28 Dec 2012 |
| 11 | 28 Dec 2012–8 Jan 2012 |
| 22 | 8 Jan 2013–30 Jan 2013 |
| 11 | 30 Jan 2013–10 Feb 2013 |
| 22 | 10 Feb 2013–4 Mar 2013 |
| 11 | 4 Mar 2013–15 Mar 2013 |
| 22 | 15 Mar 2013–6 Apr 2013 |
| 11 | 6 Apr 2013–17 Apr 2013 |
| 22 | 17 Apr 2013–9 May 2013 |
| 11 | 16 Aug 2013–27 Aug 2013 |
| 11 | 12 Nov 2013–23 Nov 2013 |
| 15 | 23 Nov 2013-8 Feb 2014 |
| 11 | 8 Feb 2014-19 Feb 2014 |
| 77 | 19 Feb 2014-7 May 2014 |
| 11 | 7 May 2014-18 May 2014 |
| 55 | 18 May 2014-12 July 2014 |
| 11 | 12 July 2014-23 July 2014 |




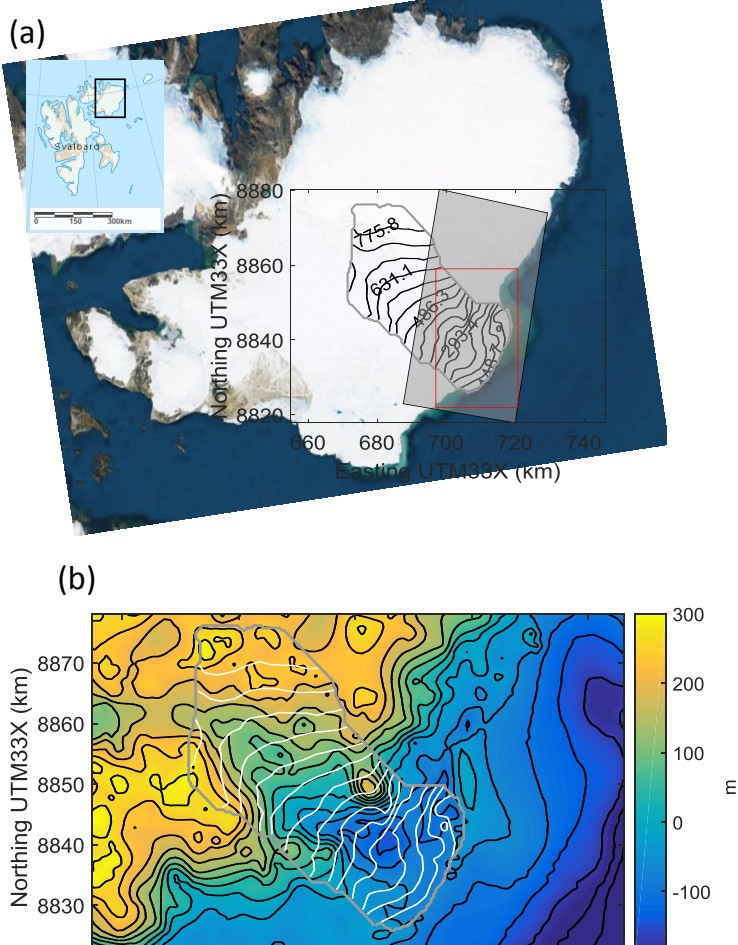

**Figure 1.** Surface and bedrock topography of Basin 3, Austfonna. (a) Surface elevation of Basin 3 contours with solid black lines (with ~48.2 m interval), on top of a satellite image of Nordaustlandet from TerraColor® Global Satellite Imagery (http://www.terracolor.net/). The gray transparent box shows the coverage of the TerraSAR-X scene (30 April 2012). The model domain of HiDEM is outlined with red box. The insert at the upper left corner shows the ice cap's location within the Svalbard archipelago; (b) Bedrock topography is color-coded, contoured with black solid line with a ~37.1 m interval and superimposed by surface elevation contours (white solid line with ~ 48.2 m interval). The gray solid line outlines Basin 3 and the model domain of Elmer/Ice in both panels.




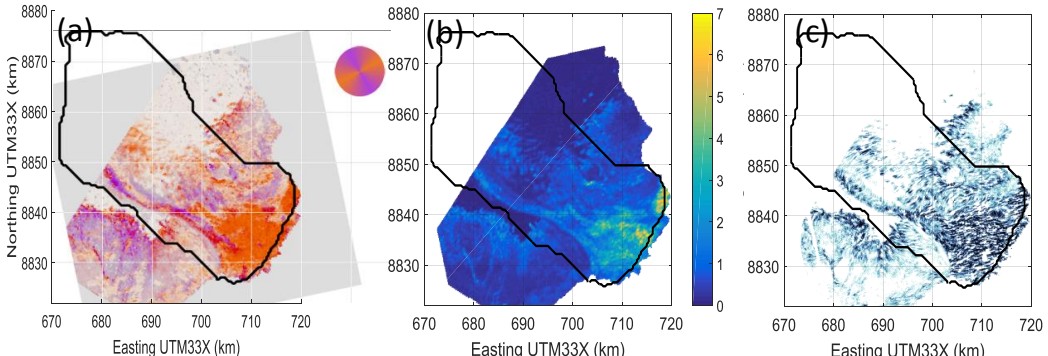

**Figure 2.** The crevasse maps created from Radon-transform. (a) The orientation of crevasses indicated by a color wheel, in which the strength of the signal controls the saturation; (b) The highest responding orientation from the Radon-transform ($\bar{s}(\theta)$) with the color bar indicating the intensity; (c) The cartographic map indicating both the orientation and intensity of the two strongest responding orientations. Basin 3 is outlined by black solid line.






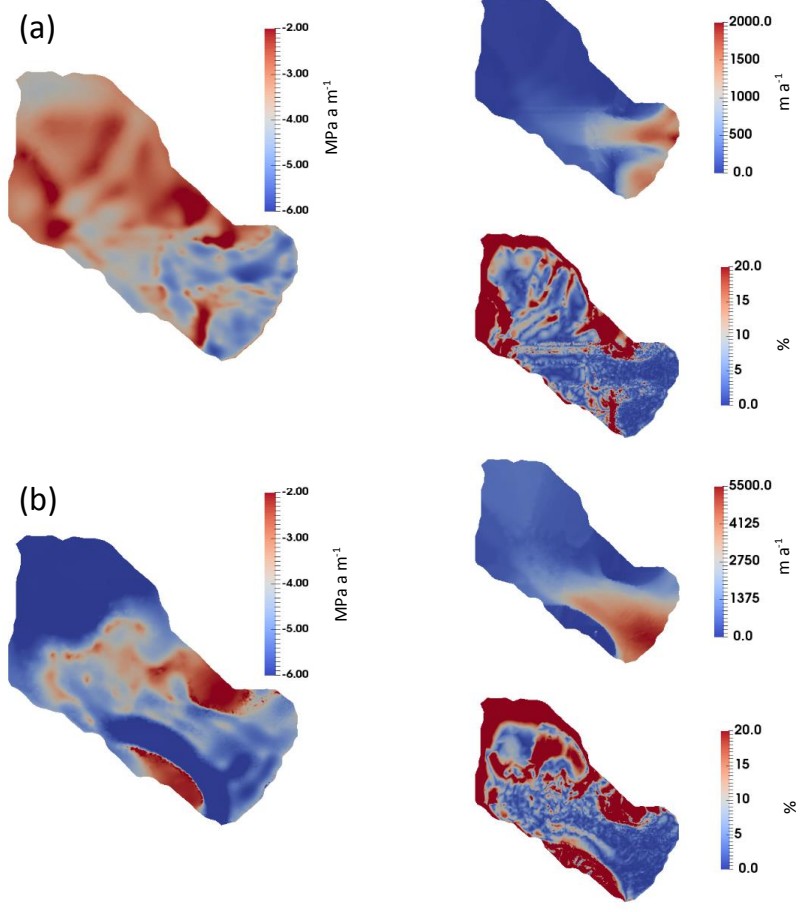

**Figure 3.** Basal friction coefficient inverted from surface velocity data in (a) 18-29 August 2012 ($C_{pre}$) and (b) 16-27 August 2013 ($C_{post}$). Both panels display basal friction coefficient shown onto the left, surface velocity data after post-processing (Sect. 2.2) shown on the upper right and the relative difference between observed and modeled surface velocity magnitude shown on the lower right.





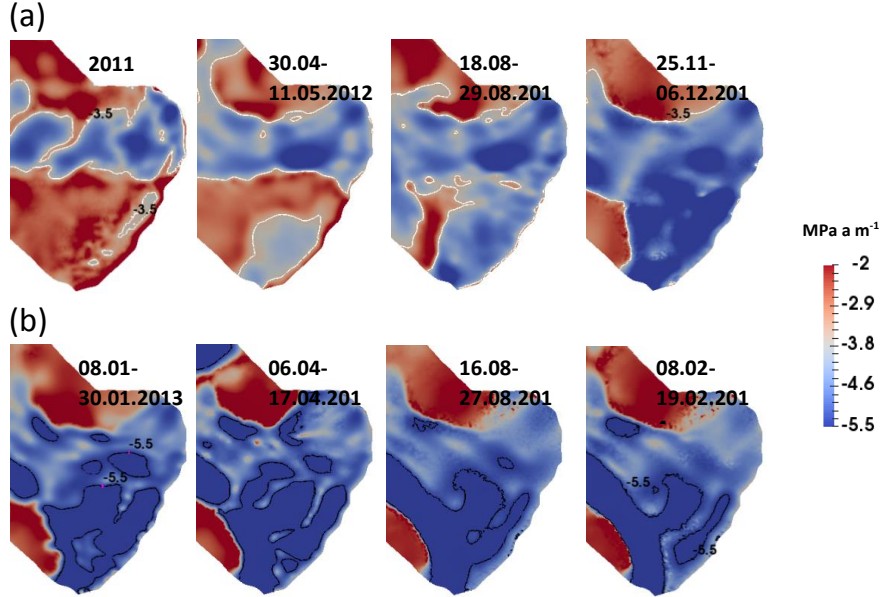

**Figure 4.** The evolution of basal friction coefficient ($C$) shown for the model domain of HiDEM. (a) $\log_{10}(C)$, overlain with white contour lines showing $\log_{10}(C) = -3.5$ (low friction), from the time before the peak of the surge; (b) $\log_{10}(C)$, overlain with black contour lines showing $\log_{10}(C) = -5.5$ (almost vanishing friction), from the time period at and after the peak of the surge.





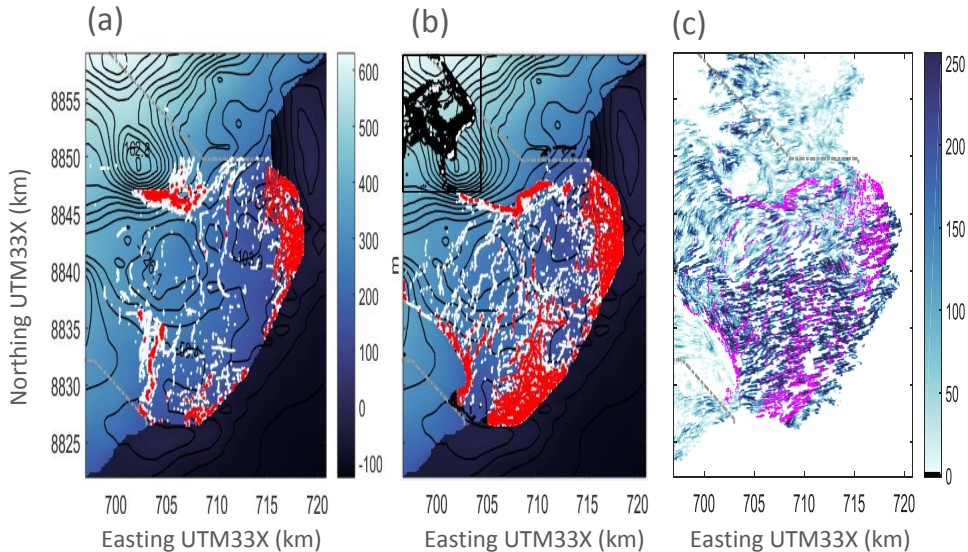

**Figure 5**. Crevasses distribution from HiDEM on (a) August 2012 and (b) August 2013 and
(c) satellite observation. The color of the underlying image in (a) and (b) shows the surface
elevation of the glacier on land. Bedrock topography contours are shown in black with a ~23.7
m interval. White dots indicate the full modeled crevasse distribution in both (a) and (b). The
superimposed red dots mark the cut-through crevasses.

The Black box in (b) marks the area in which the fractures produced due to boundary effect
(Sec. 4.2) are located. The superimposed black dots shown in (b) are eliminated from the
crevasse map as they do not fulfil the definition of a crevasse in this study.

The crevasse orientation of the satellite observation on 8 August 2013 is shown in (c) (color-
coded with detecting intensity in the background). The magenta color shows the area where
modeled and ob-served crevasse match.

The basin side boundary is outlined with gray dashed line in all the sub-plots.





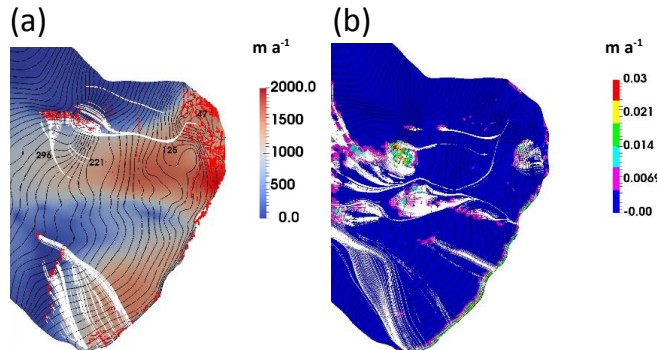

**Figure 6.** The flow paths of different water sources derived from the results in August 2012. (a) The white lines indicate the path of surface melt water after entering the basal hydrology system via cut-through crevasses (red dots) according to hydraulic potential. The modeled basal velocity magnitude is color-coded in the background. (b) The white lines indicate the water path of basal melt water from locations with in-situ melt rates above 0.005 m a$^{-1}$. The bedrock elevation is color-coded in the background. (c) The logarithm (base 10) of the basal melt rate is color-coded in the background. The colored contour lines indicate the value of the basal melt rate. The black contour lines in both (a) and (b) indicate the hydraulic potential.