# Peer review of "Simulating the roles of crevasse routing of surface water and basal friction on the surge evolution of Basin 3, Austfonna ice-cap"

_The Cryosphere, 2017_

## Referee Comment (RC1) · Anonymous Referee #1 · 6 Dec 2017

General comments:

This is an interesting study that employs several modeling approaches and a variety of remotely-sensed data sets to explore the role of water delivery through crevasses in the surge of "Basin 3" within Austfonna ice cap in Svalbard. This peculiar surge, preceded by a multi-year seasonally variable acceleration, has been studied in some detail, but the precise interplay between basal friction, crevasse formation and water delivery to the bed not thoroughly explored. The authors use the 3-D Stokes model of ice flow Elmer/Ice to invert for the basal friction field using a built-in control method, and then use this field as input to HiDEM, a discrete element model, to predict the locations

of crevasses before and after the surge. The calculated subglacial hydraulic potential distribution is used to determine the subglacial water routing, and to speculate on the mechanisms that relate surface melt to fast flow.

The scientific approach seems reasonable and the conclusions fairly well supported, though I would recommend the authors clarify some of their arguments before this paper is accepted (see below). Throughout the paper until near the end of the discussion, it was unclear to me whether the water routing to the bed through crevasses was a cause or consequence of the fast flow. The text seems to emphasize the role of water in facilitating fast flow, but the access of water to the bed via crevasses must be (at least partially) a consequence of the flow regime. I think I probably agree with the authors if they are arguing that the crevasses play an amplifying role, in that some reduction in basal traction is required to explain the formation of the crevasses that initially allow water to reach the bed. This water then accumulates in part of the domain and amplifies the acceleration of the outlet glacier. Though probably reported elsewhere, I found myself wanting to know if the thermodynamics work out: is there enough meltwater for this to be plausible given Austfonna's thermal structure? The paper would be strengthened by a clear articulation of cause versus consequence.

Specific comments (page.line):

5.157-161: I read this several times and still have difficulty understanding how this procedure provides the validation data set.

5-6: For a technical journal like The Cryosphere, I was surprised not the see the model governing equations and instead a description of the model in prose. This is perhaps a matter of personal preference, but the methodology seems less ambiguous when described with the help of equations. Early on in the model description, it should be stated that sliding is implemented and that there is some kind of thermomechanical coupling (p 6, lines 191-192). It would be useful to know a bit more about the latter without having to read Gong et al (2016).

6: It becomes clear in the results and discussion that the HiDEM simulations do not include the change in stress state resulting from pre-existing crevasses, nor the advection of crevasses. Please emphasize these points in the methods: that the crevasses predicted by HiDEM reflect the stress field at a single snapshot in time, without consideration of any pre-existing damage or advection.

7.213-215: Suggest moving this to methods or deleting.

7. section 4.2: It would be useful to know how changes in the fracture and/or bed-penetration criteria for crevasse formation affect the mismatch between modelled and observed crevasse distributions. Were these parameters set to maximize agreement, or decided upon in advance without knowledge of the outcome?

7-8: It would really help to have some annotations of the figures to orient the reader to the geographical/morphological/dynamic regions of the domain that are referenced in the text (e.g. "margin of the subglacial valley", "northern flow unit"). Perhaps a few numbers on the figures defined in the captions would do the trick.

8.244: Reword to state that the simulated and observed crevasse maps were resampled to maximize their correlation. An "appropriate resolution" would be one chosen based on the methodology and physical principles alone, rather than one chosen to maximize agreement.

8.264-272: This paragraph seems more like discussion material.

Figure 3. Separate (a) and (b) a bit better, e.g. with a line or boxes. Figure 3 is scarcely mentioned in the text (bottom of pg 6) and no description appears to be given of the 4 panels on the right-hand side. Consider adding a sentence or two of explanation to the text.

Technical corrections/queries (page.line):

The manuscript is clear and well-written overall, but still has some incorrect or awkward English phrasing. Articles (mostly "the") are missing in multiple places throughout text,

hyphens are missing or used incorrectly, sentences are started with "And" and past and present tenses and mixed. The native speaking coauthors should be able to improve this with a quick proof-reading.

2.61: Dunse et al (2015) "have"

3. 88: Suggest "Observations and data processing" as Section 2 title, since there is significant content related to methodology.

3. 99-100: "The sub-glacial hill. . ." not a sentence.

4. 113-126: paragraph needs proof-reading

5.162: Suggest "Modelling methodology" as Section 3 title, since section is related exclusively to modelling and methods for data processing were described earlier.

5.173: "brought by" => "due to"

7.227: "basin-wise" => "basin-wide"

7.236: Cannot see black box in southwest corner of Fig 5b.

8.245: "leaded" => "lead" or "led"

8.245-255: hard to know exactly where "northern", "middle", and "southern" flow units are in figures. It would really help if Figure 3c where bigger.

10.314: "factures" => "fractures"

11.356: "though" => "through"

11.376: What is meant by "cut in"? Started?

12. 386-387 "input" repeated

Figure 1. (a) Difficult to read the contour labels for surface elevation. (b) Need some labels or stated range of surface elevation contours. Need units for bedrock elevation (presumably m above sea level) more clearly stated. Sideways "m" on colorbar looks

like "E". See Figure 5 also.

Figure 2. Put (a), (b), (c) labels in a better place. Figures stretched in the vertical. Please clarify caption for panel (c). Is there a colorbar missing? It is hard to see the detail described in the text with the current size and resolution of this figure, particularly (c).

Figure 4. Some years missing on the date labels.

Figure 6. (b) "bedrock elevation is color-coded in the background"? It seems the color in (b) is described in the caption under (c), but panel (c) has been removed.

---

## Referee Comment (RC2) · Anonymous Referee #2 · 12 Jan 2018

This study utilizes remote sensing data and two numerical models to study the initiation of a surge in Basin 3, Austfonna, Svalbard. The viscous ice dynamics model Elmer/Ice is used to produce maps of the basal friction coefficient for the basin, using satellite-derived surface velocity maps over the period 2012-2014. The model states produced by these inversions are used to initialize a discrete element model for simulating the locations and patterns of crevasses in the ice. The modeled crevasse patterns are compared with observationally-derived crevasse patterns. The locations of crevasses are used to infer the sources where surface meltwater can reach the bed. Hydraulic potential maps are created to infer the potential flow pathways of this meltwater along the bed. The modeling results support the hypothesis that the surge initiation was

the result of hydro-thermodynamic feedbacks associated with summer melt, in which meltwater reaches the bed through crevasses, which then lubricates the glacier and enhances longitudinal stresses, which then promotes the formation of new crevasses which increases the catchment area for subsequent surface meltwater.

This is a timely contribution to an important topic in glaciology. Crevasses are certainly important to many feedbacks in glacier dynamics, and this study combines a model that can explicitly create crevasses with a more traditional ice dynamics model that can simulate the time-evolution of the system. The manuscript is generally well written, although it could benefit from some minor polishing for English grammar and sentence structure. The model descriptions are somewhat incomplete, such that it would not be possible to reproduce or confirm the results of this study. This can be easily addressed with additional text describing some of the explicit modelling choices.

Specific comments:

L 27: "containing a marine-terminating..."

L 70: awkward sentence, perhaps "previous crevasse modeling studies..."

L 75, 198: discrete element models are not "first principle" models unless you are explicitly modeling a particulate medium. Glaciers are not composed of idealized spheres of ice connected in a lattice framework. These are model choices used to represent a certain class of phenomena, but such a model type does not arise inherently from first principles. This is not a criticism of the model itself, but it is misleading to consider it a first-principles model.

L 87: "surge that occurred"

L 92-94: You have here a $\sim$30 year discrepancy in time between the surface elevation model and the thickness observations used to create your bedrock map. Why not use the Moholdt and Kaab DEM that you later mention on L 110?

L 99-100: not a complete sentence

Section 2.3: this seems like it belongs more in the Methods section below, as it is not really "observations"

L 166: you mention modeling ice flowing over a rigid bed, but earlier mentioned that surge behaviour may result from ice flowing over a deformable bed. Perhaps worth commenting on this here?

L 175: You mention the "slippery" terminus, but wasn't the terminus the last to mobilize when the surge initiated? Maybe I'm missing something here, or perhaps you're describing the terminus during the surge?

Inversion routine: did you add any regularization in your inversions to prevent overfitting the observations? If so, how did you decide how much? If not, why? This seems to be an important point. Regularization is commonly (and appropriately) applied in this kind of work. If you used it, you need to describe it in detail here. If not, some justification of why not is needed.

L 192-193: Ice temperatures are quite important in this kind of modeling. More description is needed here on how you computed spatially-varying temperature fields. Even if the details are in another reference, a general description of how you went about this is needed.

L 202: Elastic deformation is often considered static, that is there is no time dependence. HiDEM is not a static model, but rather a dynamic model if you are using time stepping and accounting for dynamic stress propagation. Perhaps a semantic point...

L 203: typically a modeled glacier will approach a new equilibrium... (real glaciers do not produce crevasse fields in minutes!)

L 204: See my comment above about reproducibility. A lot of choices are made when setting up a numerical model. For a study to be able to be reproduced/verified, you need to describe these choices. It's okay to refer some general description in a reference, but there is essentially no detail on the HiDEM model implementation here.

What kind of fracture criterion was used? How was the time stepping implemented? What kind of stopping criterion was used for the time stepping? How large are your discrete particles? Are they uniform in size and spacing? How sensitive are the model results to these choices? Would modifying any of these choices lead to better/worse agreement with the crevasse observations from this study?

Basal friction maps: some metric of the misfit (e.g. root mean square) would be useful to report to give an indication of the quality of the inversions. The misfit panels in Figure 3 have quite a lot of saturated regions on both the high and low ends of the misfit color scale. With these alone it is difficult to judge the quality of the fits.

It would be nice to see the evolution of velocities along with the friction evolution, for context.

L 232-233: I'm not sure what you mean by "keep" the fractures

L 235-235: the additional black region you mention is difficult to see in the figure. I'm not sure I see what you mean.

L 244: by "appropriate" you really mean you re-sampled until you got the best agreement. This is not necessarily an objective "appropriate" resolution for comparing model output with observations (smoothing crevasses over 4.5 km kind of defeats the purpose of having discrete crevasses, doesn't it?)

L 261: 60 degrees is quite a mismatch, any comment on why this is the case here? Panel 6c is mentioned in the caption of Figure 6, but not shown

L 314: "factures" → "fractures"

L 319: "emphasis" → "emphasize"

L 319-320: awkward sentence

Figure 4: some dates are cut off in the panels

[Figure]

References: check the reference list against those used in the text, it appears that a separate reference list has been concatenated to the end of the document (different font)

---

## Author Comment (AC1) · 27 Feb 2018

Dear Editor and Reviewers

We have revised the manuscript 'Simulating the roles of crevasse routing of surface water and basal friction on the surge evolution of Basin 3, Austfonna ice-cap' in response to the reviews. This includes more description of the models, clarifying the methodology of crevasses map generating from the satellite image and modeled crevasses distribution validation, a few modifications in the discussion section as well as the suggested modification for the figures (attached).

[Figure]

A point by point response to both reviewers is attached as a pdf file. We respond to the specific referee points (in bold) below. Our replies are in normal black font, original manuscript quotes are in italic, and new text in the manuscript is in blue.

Thank you for your consideration. Yongmei Gong, Thomas Zwinger, Jan Åström, Bas Altena, Thomas Schellenberger, Rupert Gladstone, John C. Moore

Please also note the supplement to this comment:
https://www.the-cryosphere-discuss.net/tc-2017-180/tc-2017-180-AC1-supplement.pdf

610

[Figure]

(a)

(b)

[Figure]

[Figure]

[Figure]

(a)   (b)   (c)

**Figure 5.** Crevasse distribution from HiDEM on (a) August 2012 and (b) August 2013 and (c) satellite observation. The color of the underlying image in (a) and (b) shows the surface elevation of the glacier. Bedrock topography contours are shown in black with a ~23.7 m interval. All the dots in both (a) and (b), regardless of the color, indicate the modeled crevasse distribution from HiDEM. The red dots are the cut-through crevasses. The red dots in the

**Supplement:**

Author's response to reviews of: Simulating the roles of crevasse routing of surface water and basal friction on the surge evolution of Basin 3, Austfonna ice-cap

Yongmei Gong, Thomas Zwinger, Jan Åström, Bas Altena, Thomas Schellenberger, Rupert Gladstone, John C. Moore

February 25, 2018

Dear Editor and Reviewers

We have revised the manuscript 'Simulating the roles of crevasse routing of surface water and basal friction on the surge evolution of Basin 3, Austfonna ice-cap' in response to the reviews. This includes more description of the models, clarifying the methodology of crevasses map generating from the satellite image and modeled crevasses distribution validation, a few modifications in the discussion section as well as the suggested modification for the figures (attached).

A point by point response to both reviewers is attached as a pdf file. We respond to the specific referee points (in **bold**) below. Our replies are in normal black font, original manuscript quotes are in *italic*, and new text in the manuscript is in blue.

Thank you for your consideration.

Yongmei Gong, Thomas Zwinger, Jan Åström, Bas Altena, Thomas Schellenberger, Rupert Gladstone, John C. Moore

**Response to anonymous referee 1**

We thank Anonymous Referee 1 for his/her thorough review of our original manuscript. The referee requested some revisions and we have in general agreed that they were needed and carried them out.

**General Comments:**

**Throughout the paper until near the end of the discussion, it was unclear to me whether the water routing to the bed through crevasses was a cause or consequence of the fast flow. The text seems to emphasize the role of water in facilitating fast flow, but the access of water to the bed via crevasses must be (at least partially) a consequence of the flow regime. I think I probably agree with the authors if they are arguing that the crevasses play an amplifying role, in that some reduction in basal traction is required to explain the formation of the crevasses that initially allow water to reach the bed. This water then accumulates in part of the domain and amplifies the acceleration of the outlet glacier. Though probably reported elsewhere, I found myself wanting to know if the thermodynamics work out: is there enough meltwater for this to be plausible given Austfonna's thermal structure? The paper would be strengthened by a clear articulation of cause versus consequence.**

Yes a good point. We realize that we need to clarify the role of the crevasses and consequentially, the surface melt water in the "hydro-thermodynamic" feedback. This corresponds both to the short

term seasonal speed up, and the role of the crevasses in the long term active surge phase. Certainly, the crevasses were initiated as a consequence of extensional flow that resulted from changes in basal thermal structure at the early stage of the active surge phase and were the triggering and enhancing factor of the 'annual hydro-thermodynamic feedback' cut in later on.

Verification of the "hydro-thermodynamic" feedback cannot be done in this study as we have neither the amount of the water reaching the bed nor a basal sliding relation engaging the basal effective pressure. However, the basal temperature distribution inversely calculated from the glacier geometry and velocity in Gong et al. (2016) has shown that the presence of a partially temperate bed in 1995 and the expansion of the temperate region from 1995 to 2011, which is consistent with the existence of basal melt water in the early stage of the surge active phase. Then calculated flow paths of both surface and basal melt water in 2012 correspond well with the fast flowing area, indicating the possible contribution of the basal water to the acceleration of the ice flow.

We modified the original text in Sec. 5:

from

*"We agree that the so-called "hydro-thermodynamic" feedback proposed by Dunse et al. (2015) could explain the development of the surge in Basin 3 in general. Based on our results we now further present arguments to emphasis the role of crevasse formation, summer melt and basal hydrology system played in the seasonal speed-up events."*

to

"We cannot directly simulate or quantify the effects of the surface melt water or basal melt water on the surge development due to the lack of a basal effective pressure dependent sliding relation. However, based on our results we can still present arguments to emphasize the role of crevasse, summer melt and basal hydrology system in the seasonal speed-up events."

and from

*"In the end, our results support the "hydro-thermodynamic' mechanism, in which crevasses provide access for surface melt water to reach the bed. We have demonstrated that cut-through crevasses are likely to be present approaching the surge in Basin-3, and that water flow paths route surface meltwater along flow paths corresponding to the regions of observed fast flow. While Dunse et al. (2015) are unspecific as to the cause of "hydro-thermodynamic" initiation zone, we propose that basal melt water, resulting from the build-up of the reservoir area and gradual thickening of ice (and hence raising of basal temperatures) during the quiescent phase, could sufficiently enhance flow speeds to initiate cut-through crevasses. Given that basal meltwater fluxes are likely to be at least an order of magnitude lower than surface meltwater or runoff fluxes, their impact on glacier sliding is likely to be much smaller. We suggest that basal meltwater, which is likely to be primarily routed toward the northern rather than southern flow unit due to topographic constraints (Fig. 6b), caused the speed up from the quiescent phase during the last part of the 20th century and early 21st century. This would require two key developments from quiescent to surge phase. Firstly, the initiation of sliding after ice thickening provided sufficient insulation for the bed to reach pressure melting temperature and generate sufficient meltwater, which could have occurred during the early nineties. Then at some point before August 2012 extensional flow due to sliding could have become sufficient to cause cut-through crevasses,*

*leading to further acceleration and the surge onset due to the annual "hydro-thermodynamic" feedback.*

*It is not clear at which point the "hydro-thermodynamic" feedback cut in, though it is likely to have first occurred in the northern flow unit, due to this unit's earlier acceleration. We suggest that the "hydro-thermodynamic" feedback cut in for the southern unit in 2011 or early 2012 due to crevasses penetrating near the southern margin (Fig. 5a), rapidly causing the basin wide surge."*

to

"Then we also discuss the role of the crevasses formation in the long term acceleration. These are initiated as a consequence of extensional flow resulting from changes in the basal thermal structure in an early post-quiescent phase and act as the triggering and enhancing factor in the so-called 'annual hydro-thermodynamic feedback' proposed by Dunse et al. (2015). While Dunse et al. (2015) are unspecific as to the cause of "hydro-thermodynamic" initiation zone in the long term glacier acceleration, we propose that basal melt water resulting from the gradual thickening of ice (raising basal temperatures) during the quiescent phase, could sufficiently enhance flow speeds to initiate cut-through crevassing. The basal temperature distribution inversely calculated from the glacier geometry and velocity (Gong et al., 2016) showed a partially temperate bed in 1995 and expansion of the temperate region from 1995 to 2011, which is consistent with the presence of water at the bed. Given that basal meltwater fluxes are likely to be at least an order of magnitude lower than surface meltwater or runoff fluxes, basal melt probably has a relatively small influence on glacier sliding. We suggest that water at the bed, which is likely to be primarily routed toward the northern rather than southern flow unit due to topographic constraints (Fig. 6b), caused the speed up from the quiescent phase during the last part of the 20th century and early 21st century.

This would require two key developments from quiescent to surge phase. Firstly, the initiation of sliding after ice thickening provided sufficient insulation for the bed to reach pressure melting temperature and generate meltwater. This could have occurred during the early nineties. Then at some point before August 2012, extensional flow due to sliding became sufficient to cause cut-through crevasses leading to further acceleration and the surge onset due to the annual "hydro-thermodynamic" feedback. We have demonstrated that cut-through crevasses are likely to be present just prior to the surge in Basin 3, and that surface meltwater can flow along the paths corresponding to the regions of observed fast flow.

It is not clear at which point the "hydro-thermodynamic" feedback cut in, though it is likely to have first occurred in the northern flow unit, due to this unit's earlier acceleration. We suggest that the "hydro-thermodynamic" feedback cut in for the southern unit in 2011 or early 2012 due to crevasses penetrating near the southern margin (Fig. 5a), rapidly causing the basin wide surge.

Direct verification of the long term evolution of the surge active phase discussed above cannot be provided without quantification of the water reaching the bed and a basal sliding relation engaging the basal effective pressure. However our approach and results can throw some light on future studies of coupled ice dynamic/thermodynamic/hydrology simulations."

**Specific comments (page.line):**

**5.157-161: I read this several times and still have difficulty understanding how this procedure provides the validation data set.**

We will need to validate the HiDEM simulation of crevasse locations by comparing them with the observational image of crevassing. This is a non-trivial exercise as the complete crevasse pattern is challenging to identify in optical imagery. Hence we used the following procedure to create an observational map of crevassing (Fig. 3c). We used both the orientation ($\theta$) of crevasse clusters and their response ($\tilde{s}(\theta)$) extracted from the Radon transformation with a line integral convolution. Later we use the Kappa statistical method to compare the similarity of the HiDEM and the observationally-based crevasse patterns.

We have modified the original text from:

*'To use the detected crevasse zones as a validation for our modeled crevasse distribution we transformed $\tilde{s}(\theta)$ and the orientation ($\theta$) into a cartographic representation (Fig. 3c). To do so, an empty image was randomly seeded with high intensities. Then a kernel with an elongated shape was convoluted over the image. This kernel was adaptive, as the orientation of the elongated shape is dependent on the orientation of the highest responding orientation signals in every window.'*

to

"We wish to compare the simulated crevasse pattern from HiDEM with these results from the observation. To identify crevasse zones and their alignment in the satellite images we process an empty image array for each 300 m×300 m window with randomly seeded high intensity values. Then a simplified line integral convolution was applied to add each element of the image to its local neighbors, weighted by a kernel. The kernel has an elongated shape. The orientation of the shape is dependent on $\theta$ at the underlying position. The response of the kernel (the intensities within) was dependent on $\tilde{s}(\theta)$ extracted from the underlying position. The resulting image is shown in Fig. 3c, and will be compared with the modeled crevasses distribution visually as well as using the statistical Kappa method."

**5-6: For a technical journal like The Cryosphere, I was surprised not the see the model governing equations and instead a description of the model in prose. This is perhaps a matter of personal preference, but the methodology seems less ambiguous when described with the help of equations. Early on in the model description, it should be stated that sliding is implemented and that there is some kind of thermomechanical coupling (p 6, lines 191-192). It would be useful to know a bit more about the latter without having to read Gong et al (2016).**

Agreed. We have modified/added the following text in Sec.3.1:

[revised manuscript text omitted]

**6: It becomes clear in the results and discussion that the HiDEM simulations do not include the change in stress state resulting from pre-existing crevasses, nor the advection of crevasses. Please emphasize these points in the methods: that the crevasses predicted by HiDEM reflect the stress field at a single snapshot in time, without consideration of any pre-existing damage or advection.**

Yes, this is worth being more explicit about. Considering that the time step size in HiDEM is $10^{-4}$ s the modeled crevasses distribution does, somewhat, reflect the stress field instantaneously. Thus we added the following texts in Sec.3.2:

L225-232: "All the simulations in this study were carried out with 30 m spatial resolution (the particles are uniformly shaped and initially uniformly spaced). We used a time step length of $10^{-4}$ s, and ran a simulation until the glacier began to approach an equilibrium state. Compared to viscous flow, elastic deformation and fracturing processes are very rapid, and a typical simulation covers about ~ 10 minutes of glacier dynamics. At the end of a simulation, a crevasse field has formed. HiDEM reflects the instantaneous stress field calculated for the time of the input boundary conditions without consideration of any pre-existing damage or advection. Further details of the model, including sensitivity of the chosen parameters to the model results are discussed in Åström et al. (2013, 2014) and Riikilä et al. (2015). All parameters were set beforehand"

**7.213-215: Suggest moving this to methods or deleting.**

We agree that the original text in L213 – L215 is unnecessary. To make the reading more natural we modified the original L213 -223: *'We investigate the evolution of basal friction using inverse modeling to determine C from the observed surface velocity between April 2012 and July 2014, spanning the period of the Basin 3 peak surge velocities in January 2013. We focus on the lower region close to the terminus that is fully covered by TSX velocity observations.*

*To make the pattern of the C distribution clearer we plotted the common logarithm of C (log10 (C)), instead of C itself. Figure 4a shows a clear expansion of low friction area (log10 (C) ⩽ -3.5) both inland and to the frontal region in the southern basin before the glacier enters the peak of the surge. In 2011 the low friction patches in the central and southern basin were disconnected from the inland region and also behind a stagnant terminus.*

*In April 2012, before the summer melt season, a low friction region also appeared in the southern corner, though still with a stagnant ice front. The low friction area of the northern flow unit slightly expanded to the south through the relatively flat frontal area. However, the fast flow did not expand*

*beyond the margin of the sub-glacial valley, which exited through the northern part of the calving front (Sect. 2.1; Fig. 4a), and might impose some restriction to the expansion of fast flow.'*

to

L281 – 289: "Figure 4 shows the friction pattern of the region that is fully covered by TSX velocity observations between April 2012 and July 2014, spanning the period of the Basin 3 peak surge velocities in January 2013. To make the pattern of the $C$ distribution clearer we plotted the common logarithm of $C$ (log10 ($C$)). Figure 4a shows a clear expansion of low friction area (log10 ($C$) $\leqslant$ -3.5) both inland and to the frontal region in the southern basin before the glacier enters the peak of the surge.

In 2011 the low friction patches in the central and southern basin were disconnected from the inland region and also lie behind a stagnant terminus. Before May 2012, the enlarged low friction area in both northern and southern glacier terminus did not expand across the flat glacier bed in between them, which might impose some topographic restriction to the expansion of the fast flow."

**7. section 4.2: It would be useful to know how changes in the fracture and/or bed penetration criteria for crevasse formation affect the mismatch between modelled and observed crevasse distributions. Were these parameters set to maximize agreement, or decided upon in advance without knowledge of the outcome?**

Agree. We have added more model description in Section 3.2

"HiDEM is a model for fracture formation and dynamics. In HiDEM, an ice body is divided into discrete particles connected by massless beams. The version of HiDEM used here is purely elastic, rather than visco-elastic (Åström et al., 2013). The elastic version is sufficient for the purpose of locating fractures governed by glacier geometry and basal friction. If the initial state of a model glacier is out of elastic equilibrium, deformation within the ice will appear as a result of Newtonian dynamics.

The explicit scheme for simulating the Newtonian dynamics and the elastic modulus can be found in Riikilä et al. (2015). We use a Young's modulus $Y = 2.0$ GPa and a Poisson ratio $\nu \approx 0.3$ for the modeled ice here. The modeled ice fractures if the stress on a beam exceed a fracture stress criterion (stretching or bending). The fracture stress is ~ 1MPa.

All the simulations in this study were carried out with 30 m spatial resolution (the particles are uniformly shaped and initially uniformly spaced). We used a time step length of $10^{-4}$ s, and ran a simulation until the glacier began to approach an equilibrium state. Compared to viscous flow, elastic deformation and fracturing processes are very rapid, and a typical simulation covers about ~ 10 minutes of glacier dynamics. At the end of a simulation, a crevasse field has formed. HiDEM reflects the instantaneous stress field calculated for the time of the input boundary conditions without consideration of any pre-existing damage or advection. Further details of the model, including sensitivity of the chosen parameters to the model results are discussed in Åström et al. (2013, 2014) and Riikilä et al. (2015). All parameters were set beforehand.

We then also modified the text in Section 4.2 to make the crevasse validation and cut through crevasses selection procedure clearer. First of all, we have double checked the modeled results from HiDEM that the width of the all the 'fractures' is larger than 0.055m. Therefore we did not actually eliminate any modeled fractures at the stage. We have changed the original L232-233:

*'We used a minimum fracture width of 0.05 m to identify a crevasse in HiDEM, which allowed us to keep most of the fractures across the whole model domain.'*

to L298:

"All the fractures calculated by HiDEM are wider than 0.055m, of which we regard as crevasses in this study."

Secondly, we compared all the modeled crevasses with the crevasses map generated from the satellite image not only the cut-through ones. We moved the texts originally in L237- 239:

*'We defined cut-through crevasses as crevasses that penetrate through 2/3 ice depth and assume that they could cut through the full depth of ice if filled with water and potentially route surface melt water into the basal hydrology system vertically.'*

to Section4.3 Surface and basal water sources (L339 - 340):

"We defined cut-through crevasses as crevasses that penetrate through 2/3 ice-depth and assume that they could cut through the full depth of ice if filled with water and potentially route surface melt water into the basal hydrology system vertically."

As suggested by the reviewer we also checked the Kappa coefficient when including the artificial crevasses. The $4.6 \times 4.6$ km smoothing window for re-sampling is used to maximize the agreement. We think visual comparison can judge the agreement. The statistical method is just used to give the reader the quantitative information. Thus we changed the original L240 – 250: *'The crevasse distribution from $C_{post}$ was validated using the crevasse map generated from satellite observations acquired on 240 4th August 2013. The cartographic map of the crevasse detection (Fig. 3c) from the satellite observation was used for the validation. To estimate the statistical quality of the simulated crevasse field with the observationally estimated map we calculated the Kappa coefficient (K) (Wang et al., 2016). As almost any two maps will be significantly different with large sample size (> 62483) (Monserud and Leemans, 1992), we firstly re-sampled the two maps to an appropriate resolution. Experimentation leaded us to require a 4.6×4.6 km smoothing window to achieve substantial agreement (K = 0.71) (Cohen, 245 1960) between the maps. At higher resolutions K is worse for a variety of reasons: the ice dynamics model cannot advect crevasses, hence many crevasses in the image that in reality were created further upstream were simply not present in the simulation; crevasse densities are very variable and even at 1.5 km resolution the distribution is not smooth (K = 0.45); and the observationally derived map is not a perfect representation of reality. We next discuss the crevasse patterns derived from observations and those from the discrete element model in detail.'*

to

L317 – 326: "Although the visual comparison between the two maps shows a general agreement (Fig 5c), estimation of statistical quality of the simulated crevasse field with the observationally estimated map is necessary. We calculated the Kappa coefficient (K) (Wang et al., 2016) to quantify the agreement, but this is not trivial as almost any two maps will be significantly different with large sample size (> 62483) (Monserun and Leemans, 1992). We achieve moderate agreement (Cohen, 1960), (K = 0.45) when re-sampling the two maps with a $1.5 \times 1.5$ km smoothing window and substantial agreement (K = 0.71) with a $4.6 \times 4.6$ km smoothing window. When including the artificial crevasses (defined at the beginning of the section) the agreement is only fair (K ~= 0.30) for both re-sample windows. A variety of reasons can explain the resolution dependency of the results of the Kappa method: the ice dynamics model cannot advect crevasses, hence many

crevasses in the image that in reality were created further upstream were simply not present in the simulation; crevasse densities are very variable; and the observationally derived map is not a perfect representation of reality."

**7-8: It would really help to have some annotations of the figures to orient the reader to the geographical/morphological/dynamic regions of the domain that are referenced in the text (e.g. "margin of the subglacial valley", "northern flow unit"). Perhaps a few numbers on the figures defined in the captions would do the trick.**

Agreed. We marked SV (sub-glacial valley), OD (over-deepening area of the bed), NF (Northern flow unit) and SF (Southern flow unit) on Fig. 1b and added the following texts in the caption:

''SV' marks the subglacial valley that runs between two bedrock maxima in the northeast and southwest and extends several tens of kilometers upstream and downstream. 'OD' marks the minimum bedrock height for Basin 3 and is within an over-deepening in the lower part of the valley. 'NF' marks the downstream area of the northern flow unit of the glacier, which runs from the upstream of the valley and exits from the northern terminus. The alignment of these labels roughly indicates the flow direction. Similarly, 'SF' marks the downstream area of the southern flow unit.'

In order to orient the readers to what we mean by saying 'the crevasses above the margins of the sub-glacial valley' we added two yellow boxes in Fig. 5a and the following text in the capitation:

"The red dots in the yellow boxes in (a) are referred to as cut-through crevasses above the sub-glacial valley margins and are used for calculating the flow paths of the surface melt reached the bed."

**8.244: Reword to state that the simulated and observed crevasse maps were resampled to maximize their correlation. An "appropriate resolution" would be one chosen based on the methodology and physical principles alone, rather than one chosen to maximize agreement.**

Agreed. We think visual comparison is more sufficient to judge the agreement. The statistical method is just used to give the reader the quantitative information. Thus we changed the original L240 – 250 from: '*The crevasse distribution from $C_{post}$ was validated using the crevasse map generated from satellite observations acquired on 240 4th August 2013. The cartographic map of the crevasse detection (Fig. 3c) from the satellite observation was used for the validation. To estimate the statistical quality of the simulated crevasse field with the observationally estimated map we calculated the Kappa coefficient (K) (Wang et al., 2016). As almost any two maps will be significantly different with large sample size (> 62483) (Monserud and Leemans, 1992), we firstly re-sampled the two maps to an appropriate resolution. Experimentation leaded us to require a 4.6×4.6 km smoothing window to achieve substantial agreement (K = 0.71) (Cohen, 245 1960) between the maps. At higher resolutions K is worse for a variety of reasons: the ice dynamics model cannot advect crevasses, hence many crevasses in the image that in reality were created further upstream were simply not present in the simulation; crevasse densities are very variable and even at 1.5 km resolution the distribution is not smooth (K = 0.45); and the observationally derived map is not a perfect representation of reality. We next discuss the crevasse patterns derived from observations and those from the discrete element model in detail.*'

to the texts below and put the them after the visual comparison:

L317 – 326 "Although the visual comparison between the two maps shows a general agreement (Fig 5c), estimation of statistical quality of the simulated crevasse field with the observationally

estimated map is necessary. We calculated the Kappa coefficient (K) (Wang et al., 2016) to quantify the agreement, but this is not trivial as almost any two maps will be significantly different with large sample size (> 62483) (Monserun and Leemans, 1992). We achieve moderate agreement (Cohen, 1960), (K = 0.45) when re-sampling the two maps with a $1.5 \times 1.5$ km smoothing window and substantial agreement (K = 0.71) with a $4.6 \times 4.6$ km smoothing window. When including the artificial crevasses (defined at the beginning of the section) the agreement is only fair (K ~= 0.30) for both re-sample windows. A variety of reasons can explain the resolution dependency of the results of the Kappa method: the ice dynamics model cannot advect crevasses, hence many crevasses in the image that in reality were created further upstream were simply not present in the simulation; crevasse densities are very variable; and the observationally derived map is not a perfect representation of reality."

**8.264-272: This paragraph seems more like discussion material.**

Agreed. We modified and moved the original L264 – 272 in Sec. 4.2 Crevasses Distribution and Validation from: '*This mismatch of the orientation between the modeled and observationally derived crevasse distribution in the middle upper area (Fig. 5c) may be due to HiDEM only simulating the ad-hoc formation and not advection of crevasses, thus no crevasse 265 formation history can be inferred from the model. The inclusion of crevasse advection could be implemented in a two-way coupling of HiDEM with a continuum model in future studies. The mismatch of the crevasse density (Fig. 5c) at the northern and southern frontal area could be caused by the mismatch of ice front position between the reality and the model. Although in reality the ice front advanced for several kilometers after the full-surge, it was kept fixed in position in Elmer/Ice (Sect. 3.1). The shape and steepness of the ice front likely affects the behavior of the discrete element model. However, as they are 270 concentrated at the terminus of the glacier, these crevasses are less likely to affect the basal hydrology system on a wider scale.*'

to L372 – 379, Section 5 Discussion : 'However there is a mismatch of the orientation in the middle upper area (Fig. 5c). It may be due to that HiDEM only simulates the ad-hoc formation but not the advection of crevasses, thus no crevasse formation history can be inferred from the model. The inclusion of crevasse advection could be implemented in a two-way coupling of HiDEM with a continuum model accounting for damage transport in future studies. The mismatch of the crevasse density (Fig. 5c) at the northern and southern frontal area could be caused by the mismatch of ice front position between reality and the model. Although in reality the ice front advanced for several kilometers after the full-surge, it was kept fixed in position in Elmer/Ice (Sect. 3.1). The shape and steepness of the ice front likely affects the behavior of the discrete element model. However, as they are concentrated at the terminus of the glacier, these crevasses are less likely to affect the basal hydrology system on a wider scale.'

**Figure 3. Separate (a) and (b) a bit better, e.g. with a line or boxes. Figure 3 is scarcely mentioned in the text (bottom of pg 6) and no description appears to be given of the 4 panels on the right-hand side. Consider adding a sentence or two of explanation to the text.**

Agreed. The former Fig.3 is now Fig.2. We added two rectangle frame in Fig. 2 to separate the results from August 2012 and August 2013. We also added more description of the figure in L276-280: "Figure 2 shows that the relative errors between the modeled and observed surface velocity magnitude for both the 18-29 August 2012 and the 16-27 August 2013 snapshots are the lowest over the fast flowing region (< 5%) (Fig. 2), the areas mostly moving by basal sliding. The root-mean-squared difference of the modeled surface velocity magnitude fields in the TXS data covered

region (Fig. 1) for these two time periods are 65.0 and 190.9 m a$^{-1}$, respectively. As we are mostly interested in the ice dynamics of the fast flowing area, these errors are acceptable for the crevasse formation simulations."

**Technical corrections/queries (page.line):**

**The manuscript is clear and well-written overall, but still has some incorrect or awkward**

**English phrasing. Articles (mostly "the") are missing in multiple places throughout text,**

Thanks for pointing out. We have checked the language once again.

**Response to anonymous referee 2**

We thank Anonymous Referee 2 for their thorough review of our original manuscript. The referee suggested a number of modifications to the manuscript, and we have followed the advice given in the main.

*General Comments:*

**The manuscript is generally well written, although it could benefit from some minor polishing for English grammar and sentence structure. The model descriptions are somewhat incomplete, such that it would not be possible to reproduce or confirm the results of this study. This can be easily addressed with additional text describing some of the explicit modelling choices.**

We have checked the language once again and add more model descriptions in Section 3:

[revised manuscript text omitted]

*Specific comments:*

**L 27: "containing a marine-terminating..."**

We have changed the original sentence '*containing marine-terminating…*' to 'containing a marine-terminating…'

**L 70: awkward sentence, perhaps "previous crevasse modeling studies..."**

We have changed the original sentence '*Previous studies of modeling crevasse simulate…*' to 'Previous crevasse modeling studies simulate…'

**L 75, 198: discrete element models are not "first principle" models unless you are explicitly modeling a particulate medium. Glaciers are not composed of idealized spheres of ice connected in a lattice framework. These are model choices used to represent a certain class of phenomena, but such a model type does not arise inherently from first principles. This is not a criticism of the model itself, but it is misleading to consider it a first-principles model.**

Agree. We have deleted 'first principle' from both sentences.

**L 87: "surge that occurred"**

We have changed the original sentence from '*…for the surge occurred in Basin 3*' to '…for the surge that occurred in Basin 3.'

**L 92-94: You have here a ~30 year discrepancy in time between the surface elevation model and the thickness observations used to create your bedrock map. Why not use the Moholdt and Kaab DEM that you later mention on L 110?**

The bedrock map is not created using the surface elevation data acquired during July 2010 – December 2012. It was made by subtracting ice thickness (RES data from 1983 supplemented by two other data from 2008) from an older DEM (based on a Norwegian Polar Institute map published in 1998 and and InSAR data of Aust-fonna acquired in 1995–96. Its validity is discussed in Dunse., 2011. The bedrock elevation is used in other studies in addition to ours. In this study we assume bedrock elevation would not change in the time scale of decades then just simply updated the surface elevation to the data acquired during July 2010 – December 2012.

We realized that there is also a miswriting of the date when the older DEM was made. Thus we have changed the original text from

'*Surface elevation was derived from Cryosat altimetry data acquired during July 2010 – December 2012 (McMillan et al., 2014). McMillan et al., (2014) grouped measurements acquired over a succession of orbit cycles that are within 2-5 km2 geographic regions. Bedrock elevation (Dunse, 2011) was derived by point-wise subtracting the measured ice thickness from a 250 m resolution surface elevation that is based on the Norwegian Polar Institute (NPI) 1:250 000 topographic maps derived from aerial photography over Austfonna in 1983. The ice thickness used for generating bedrock elevation was based on airborne radio echo sounding (RES), (Dowdeswell et al., 1986) supplemented with two RES data sets from 2008 (Vasilenko et al., 2009). Marine bathymetry (2 km horizontal resolution) was from the International Bathymetry Chart of the Arctic Ocean, Version 2.0 (Jakobsson et al., 2008). Bathymetry and inland bedrock elevation were combined by using an interactive gridding scheme to eliminate the mismatch along the southern and northwest coast line (Dunse, 2011).*'

to

"Bedrock elevation (Dunse, 2011) was derived by point-wise subtracting the measured ice thickness from a 250 m resolution surface elevation that is derived from a Norwegian Polar Institute (NPI) 1:250 000 topographic maps published in 1998 and InSAR data of Austfonna acquired in 1995-96 (Unwin and Wingham, 1997). The ice thickness used for generating bedrock elevation was based on airborne radio echo sounding (RES), (Dowdeswell et al., 1986) supplemented with two RES data sets from 2008 (Vasilenko et al., 2009). Marine bathymetry (2

km horizontal resolution) was from the International Bathymetry Chart of the Arctic Ocean, Version 2.0 (Jakobsson et al., 2008). Bathymetry and inland bedrock elevation were combined by using an interactive gridding scheme to eliminate the mismatch along the southern and northwest coast line (Dunse, 2011). We assume that bedrock elevation does not have any significant changes over decadal time scales, and use it with a set of updated surface elevation data. The surface elevation was derived from Cryosat altimetry data acquired during July 2010 – December 2012 (McMillan et al., 2014). McMillan et al. (2014) grouped measurements acquired over a succession of orbit cycles that are within 2-5 km$^2$ geographic regions."

**L 99-100: not a complete sentence**

We have changed the original sentence from '*The sub-glacial hill located at roughly 700 km E and 8850 km N rising to about 250 m a.s.l.*' to 'The sub-glacial hill located at roughly 700 km E and 8850 km N rises to about 250 m above sea level.'

**Section 2.3: this seems like it belongs more in the Methods section below, as it is not really "observations"**

Agreed. We have moved the former 'Section 2.3 Crevasse map' to 'Section 3.3 Crevasse map' in 'Section 3 Methodology' and changed the order of the cited figures accordingly.

**L 166: you mention modeling ice flowing over a rigid bed, but earlier mentioned that surge behaviour may result from ice flowing over a deformable bed. Perhaps worth commenting on this here?**

As introduced in Sect.1, ideally, a soft-bed sliding mechanism needs to be simulated to be able to capture the surging behavior. However, as the main goal of this study is only to find a model approach to locate the surface melt water input sources, a linear basal sliding relation solved with an inverted parameter (C) which reflects the observation quite well (Fig. 2) is good enough to serve the purpose.

**L 175: You mention the "slippery" terminus, but wasn't the terminus the last to mobilize when the surge initiated? Maybe I'm missing something here, or perhaps you're describing the terminus during the surge?**

We meant to say that, in the present study, the fixed calving front criterion does not affect the basal shear stress at the ice front very much as the stagnant ice front has already disappeared at the time for which the simulations are carried out. We have changed the original sentence from '*On the other hand the basal shear stress calculation at the ice terminus will be affected. However the glacier bed is already very 'slippery' at the ice terminus.*' to 'The fixed calving front criterion would not distort the basal shear stress calculation at the ice terminus neither, as the basal resistance there is already low in 2012.'

**Inversion routine: did you add any regularization in your inversions to prevent overfitting the observations? If so, how did you decide how much? If not, why? This seems to be an important point. Regularization is commonly (and appropriately) applied in this kind of work. If you used it, you need to describe it in detail here. If not, some justification of why not is needed.**

Yes we did have regularization. The following sentences have been added in Section3.1

"A Tikhonov regularization term penalizing the spatial first derivatives of α is used to avoid over fitting:

$$J_{reg} = \frac{1}{2} \int_{\Gamma_b} \left(\frac{\partial \alpha}{\partial x}\right)^2 + \left(\frac{\partial \alpha}{\partial y}\right)^2 d\Gamma, \tag{10}$$

such that the total cost function is now written as:

$$J_{tot} = J_0 + \lambda J_{reg}, \tag{11}$$

where $\lambda$ is a positive ad-hoc parameter. We adopted the same procedure as in Gillet-Chaulet et al. (2012) to find the optimal $\lambda$ value."

**L 192-193: Ice temperatures are quite important in this kind of modeling. More description is needed here on how you computed spatially-varying temperature fields. Even if the details are in another reference, a general description of how you went about this is needed.**

Agreed. We added the following sentences in Sect.3.1

'The temperature distribution is calculated according to the general balance equation of internal energy written as:

$$\rho_i c_v \left(\frac{\partial T}{\partial t} + u \cdot \nabla T\right) = \nabla \cdot (\kappa \nabla T) + D{:}\sigma, \tag{12}$$

where $\kappa = \kappa(T)$ and $c_v = c_v(T)$ are the heat conductivity and specific heat of ice, respectively. $D{:}\sigma$ represents the amount of energy produced by dice deformation. The upper value of the temperature $T$ is constrained by the pressure melting point $T_m$ of ice.

The Dirichlet boundary condition at the upper surface, $T_{surf}$, is prescribed as:

$$T_{surf} = T_{sea} + \Gamma z_s, \tag{13}$$

where $T_{surf}$ is the surface ice temperature, $T_{sea} = -7.68$ °C is the mean annual air temperature at sea level estimated from two weather stations on Austfonna during 2004 and 2008 (Schuler et al., 2014) and four weather stations on Vestfonna during 2008 and 2009 (Möller et al., 2011), $\Gamma$ is = 0.004 K m$^{-1}$ is the lapse rate (Schuler et al., 2007).

An initial guess of the ice temperature, $T_{init}$, is given by:

$$T_{init} = T_{surf} + \frac{q_{geo}}{\kappa} d, \tag{14}$$

where $q_{geo} = 40.0$ mW m$^{-2}$ is the geothermal heat flux (Dunse et al., 2011) and $d$ the distance from the upper surface.

Spatially varied ice temperatures ($T$) snapshots in the flow solution were accommodated using an iterative process which includes four parts: i) Invert $C_{invert}$ for the first time with either an initial guess of $C_{init}$ and $T_{init}$ or the previously inverted $C_{prev}$ and $T_{prev}$; ii) Carry out steady state simulation for only thermodynamics to calculate $T_{invert}$ using the velocities obtained from the inversion; iii) Do the inversion again using $C_{invert}$ and $T_{invert}$ derived from the previous simulations; iv) Repeat the iteration until the differences in $C_{invert}$ and $T_{invert}$ between two successive iterations fall below a given threshold. More details about the interactive process can be found in Gong et al. (2016)."

**L 202: Elastic deformation is often considered static that is there is no time dependence. HiDEM is not a static model, but rather a dynamic model if you are using time stepping and accounting for dynamic stress propagation. Perhaps a semantic point...**

'Elastic' is used as meaning reversible deformation - in opposite to e.g. 'Viscous' or 'Fractured'. Solutions in HiDEM naturally are computed transient.

**L 203: typically a modeled glacier will approach a new equilibrium. . . (real glaciers do not produce crevasse fields in minutes!)**

Crack propagation in ice is of the order of magnitude of a few 100m/sec - which means that a crevasse field can form in a few minutes.

**L 204: See my comment above about reproducibility. A lot of choices are made when setting up a numerical model. For a study to be able to be reproduced/verified, you need to describe these choices. It's okay to refer some general description in a reference, but there is essentially no detail on the HiDEM model implementation here. What kind of fracture criterion was used? How was the time stepping implemented? What kind of stopping criterion was used for the time stepping? How large are your discrete particles? Are they uniform in size and spacing? How sensitive are the model results to these choices? Would modifying any of these choices lead to better/worse agreement with the crevasse observations from this study?**

Agreed. We have added more model descriptions in Section3.2

"**3.2 Crevasse distribution calculation by a discrete element model**

HiDEM is a model for fracture formation and dynamics. In HiDEM, an ice body is divided into discrete particles connected by massless beams. The version of HiDEM used here is purely elastic, rather than visco-elastic (Åström et al., 2013). The elastic version is sufficient for the purposes of locating fractures governed by glacier geometry boundary and basal friction. If the initial state of a model glacier is out of elastic equilibrium, deformation within the ice will appear as a result of Newtonian dynamics.

The explicit scheme for simulating the Newtonian dynamics and the elastic modulus can be found in Riikilä et al. (2015). We use Young's modulus $Y = 2.0$ GPa and the Poisson ratio $v \approx 0.3$ for the modeled ice here. The modeled ice fractures if the stress on a beam exceed a fracture stress criterion (stretching or bending). The fracture stress is $\sim$ 1MPa.

All the simulations in this study were carried out with 30 m spatial resolution (the particles are uniformly shaped and initially uniformly spaced). We used a time step length of $10^{-4}$ sec, and ran a simulation until the glacier began to approach an equilibrium state. Compared to viscous deformation, elastic deformations are very rapid, and a typical typically simulation lasted $\sim$ 10 minutes of simulated dynamics. At the end of a simulation, a crevasse field were formed. HiDEM reflect the instantaneous stress field calculated for the time of the input boundary conditions without consideration of any pre-existing damage or advection. Further details of the model, including sensitivity of the chosen parameters to the model results are discussed in Åström et al. (2013, 2014) and Riikilä et al. (2015). All parameters were set beforehand.**"

**Basal friction maps: some metric of the misfit (e.g. root mean square) would be useful to report to give an indication of the quality of the inversions. The misfit panels in Figure 3 have quite a lot of saturated regions on both the high and low ends of the misfit color scale. With these alone it is difficult to judge the quality of the fits.**

Agreed. Most of the saturated regions on the high end are slow flowing regions. The relative errors in the fast flowing area are mostly below 5%. We also calculated the RMSD and added one paragraph at the beginning of the 'Sec. 4.1 Basal friction evolution:

"Figure 2 shows that the relative errors between the modeled and observed surface velocity magnitude for both the 18-29 August 2012 and the 16-27 August 2013 snapshots are lowest over the fast flowing region ($<5\%$) (Fig. 2), the areas mostly moving by basal sliding. The root-mean-squared difference of the modeled surface velocity magnitude fields in the TXS data covered region (Fig. 1) for these two time periods are 65.0 and 190.9 m a$^{-1}$, respectively. As we are mostly interested in the ice dynamics of the fast flowing area, these errors are acceptable for the crevasse formation simulations.

**It would be nice to see the evolution of velocities along with the friction evolution, for context.**

Agreed. We have added observed speed snapshots in Figure 4.

**L 232-233: I'm not sure what you mean by "keep" the fractures**

We have double checked the modeled results from HiDEM that the width of the all the 'fractures' is larger than 0.055m. Therefore we did not actually eliminate any modeled fractures. We changed L232-233:

*'We used a minimum fracture width of 0.05 m to identify a crevasse in HiDEM, which allowed us to keep most of the fractures across the whole model domain.'*

to L298:

"All the fractures calculated by HiDEM are wider than 0.055m, which we regard as crevasses in this study."

**L 235-235: the additional black region you mention is difficult to see in the figure. I'm not sure I see what you mean.**

We realized that the description for the 'black dots' or rather all the dots in either the text or the caption of Fig.5 are quite ambiguous. Thus we have changed the original text from:

*'Many fractures were generated upstream of the sub-glacial hill (the area inside the black box in Fig. 5b); these were caused by boundary effects due to the limited domain and are excluded from the study (illustrated by 235 the region in Fig. 1a). A similar boundary effect causes incorrect crevassing in the southwest corner of the domain (also marked in black in Fig. 5b).'*

to

"The fractures marked with black dots (Fig. 5b; in both upper left and lower left corner of the domain) are generated by boundary effects due to the limited domain. Although they might be deep enough to cut through the full depth of the ice we regard them as artificial crevasses. They are irrelevant to the water routing and surge processes we focus on in this paper thus are exclude from the comparison in this section and the water routing calculation in Sect. 4.3"

and also change the caption from:

*'Crevasses distribution from HiDEM on (a) August 2012 and (b) August 2013 and (c) satellite observation. The color of the underlying image in (a) and (b) shows the surface elevation of the glacier on land. Bedrock topography contours are shown in black with a ~23.7 m interval. White dots indicate the full modeled crevasse distribution in both (a) and (b). The superimposed red dots mark the cut-through crevasses. The Black box in (b) marks the area in which the fractures produced due to boundary effect (Sec. 4.2) are located. The superimposed black dots shown in (b) are eliminated from the crevasse map as they do not fulfil the definition of a crevasse in this study.*

*The crevasse orientation of the satellite observation on 8 August 2013 is shown in (c) (colorcoded with detecting intensity in the background). The magenta color shows the area where modeled and observed crevasse match. The basin side boundary is outlined with gray dashed line in all the sub-plots.'*

to

'Figure 5. Crevasse distribution from HiDEM on (a) August 2012 and (b) August 2013 and (c) satellite observation. The color of the underlying image in (a) and (b) shows the surface elevation of the glacier. Bedrock topography contours are shown in black with a ~23.7 m interval. All the dots in both (a) and (b), regardless of the color, indicate the modeled crevasse distribution from HiDEM. The red dots are the cut-through crevasses. The red dots in the yellow boxes in (a) are the ones referred as cut-through crevasses above the sub-glacial valley margins and are used for calculating the flow paths of the surface melt reached the bed. The black dots in (b) (upper left and lower left corner) mark crevasses produced due to boundary effects in the model (Sect. 4.2). They are eliminated from the crevasse map. The rest of the crevasses are marked with white dots, and are mostly shallow crevasses, hence irrelevant to water routing. The cartographic representation of the observed crevasse orientation on 8 August 2013 is shown in (c) (color-coded with detecting intensity in the background). The magenta color shows the area where modeled and observed crevasse match.  The basin side boundary is outlined with gray dashed line in all the sub-plots.'

**L 244: by "appropriate" you really mean you re-sampled until you got the best agreement.**

**This is not necessarily an objective "appropriate" resolution for comparing model output with observations (smoothing crevasses over 4.5 km kind of defeats the purpose of having discrete crevasses, doesn't it?)**

Agreed. We think visual comparison is more sufficient to judge the agreement. The statistical method is just used to give the reader the quantitative information. Thus we changed the original L240 – 250 from: '*The crevasse distribution from $C_{post}$ was validated using the crevasse map generated from satellite observations acquired on 240 4th August 2013. The cartographic map of the crevasse detection (Fig. 3c) from the satellite observation was used for the validation. To estimate the statistical quality of the simulated crevasse field with the observationally estimated map we calculated the Kappa coefficient (K) (Wang et al., 2016). As almost any two maps will be significantly different with large sample size (> 62483) (Monserud and Leemans, 1992), we firstly re-sampled the two maps to an appropriate resolution. Experimentation leaded us to require a 4.6×4.6 km smoothing window to achieve substantial agreement (K = 0.71) (Cohen, 245 1960) between the maps. At higher resolutions K is worse for a variety of reasons: the ice dynamics model cannot advect crevasses, hence many crevasses in the image that in reality were created further upstream were simply not present in the simulation; crevasse densities are very variable and even at 1.5 km resolution the distribution is not smooth (K = 0.45); and the observationally derived map is not a perfect representation of reality. We next discuss the crevasse patterns derived from observations and those from the discrete element model in detail.'*

to the texts below and put the them after the visual comparison:

L317 – 326: "Although the visual comparison between the two maps shows a general agreement (Fig 5c), estimation of statistical quality of the simulated crevasse field with the observationally estimated map is necessary. We calculated the Kappa coefficient (K) (Wang et al., 2016) to quantify the agreement, but this is not trivial as almost any two maps will be significantly different

with large sample size (> 62483) (Monserun and Leemans, 1992). We achieve moderate agreement (Cohen, 1960), (K = 0.45) when re-sampling the two maps with a $1.5 \times 1.5$ km smoothing window and substantial agreement (K = 0.71) with a $4.6 \times 4.6$ km smoothing window. When including the artificial crevasses (defined at the beginning of the section) the agreement is only fair (K ~= 0.30) for both re-sample windows. A variety of reasons can explain the resolution dependency of the results of the Kappa method: the ice dynamics model cannot advect crevasses, hence many crevasses in the image that in reality were created further upstream were simply not present in the simulation; crevasse densities are very variable; and the observationally derived map is not a perfect representation of reality."

**L 261: 60 degrees is quite a mismatch, any comment on why this is the case here?**

One explanation could be the modeled crevasses in HiDEM only reflects a material failure due the 'instantaneous' stress field which is dominated by extensional stress corresponding to the flow direction. But in reality there could be crevasses advected from upstream and got distorted on their way downstream, especially for the shallow crevasses in the middle upper area.

These are also discussed in discussion section, L373 – L375:

'It may be due to that HiDEM only simulates the ad-hoc formation but not the advection of crevasses, thus no crevasse formation history can be inferred from the model. The inclusion of crevasse advection could be implemented in a two-way coupling of HiDEM with a continuum model accounting for damage transport in future studies.'

**Panel 6c is mentioned in the caption of Figure 6, but not shown**

We have changed the original caption from '*Figure 6. The flow paths of different water sources derived from the results in August 2012. (a) The white lines indicate the path of surface melt water after entering the basal hydrology system via cut-through crevasses (red dots) according to hydraulic potential. The modeled basal velocity magnitude is color-coded in the background. (b) The white lines indicate the water path of basal melt water from locations with in-situ melt rates above 0.005 m a-1. The bedrock elevation is color-coded in the background. (c) The logarithm (base 10) of the basal melt rate is colorcoded in the background. The colored contour lines indicate the value of the basal melt rate. The black contour lines in both (a) and (b) indicate the hydraulic potential.*'

to "Figure 6. The flow paths of different water sources derived from the results in August 2012. (a) The white lines indicate the path of surface melt water after entering the basal hydrology system via cut-through crevasses (red dots) according to hydraulic potential. The modeled basal velocity magnitude is color-coded in the background. (b) The white lines indicate the water path of basal melt water from locations with in-situ melt rates above 0.005 m a$^{-1}$. The colored contour lines indicate the value of the basal melt rate. The black contour lines in both (a) and (b) indicate the hydraulic potential."

**L 314:"factures" → "fractures"**

The original sentence has been changed from '*We used the discrete element model – HiDEM (Åström et al., 2014) to locate the possible location of crevasse factures that…*' to 'We used the discrete element model – HiDEM (Åström et al., 2014) to locate the possible location of crevasses.'

**L 319:"emphasis" → "emphasize"**

Corrected.

**L 319-320: awkward sentence**

The original paragraph has been changed from '*We agree that the so-called "hydro-thermodynamic" feedback proposed by Dunse et al. (2015) could explain the development of the surge in Basin 3 in general. Based on our results we now further present arguments to emphasis the role of crevasse formation, summer melt and basal hydrology system played in the seasonal speed-up events.*'

to "We cannot directly simulate or quantify the effects of the surface melt water or basal melt water on the surge development due to the lack of a basal effective pressure dependent sliding relation. However, based on our results we can still present arguments to emphasize the role of crevasse, summer melt and basal hydrology system in the seasonal speed-up events."

**Figure 4: some dates are cut off in the panels**

We have fixed the problem.

*References: check the reference list against those used in the text, it appears that a separate reference list has been concatenated to the end of the document (different font)*
We have fixed the problem.